# Towards Reliable and Efficient Backdoor Trigger Inversion via Decoupling Benign Features

**Xiong Xu**[1,*], **Kunzhe Huang**[3,*], **Yiming Li**[1,2,*,†], **Zhan Qin**[1,†], **Kui Ren**[1]
[1]The State Key Laboratory of Blockchain and Data Security, Zhejiang University
[2]ZJU-Hangzhou Global Scientific and Technological Innovation Center
[3]Alibaba Cloud
{xiongx, li-ym, zhanqin, kuiren}@zju.edu.cn; huangkunzhe.hkz@alibaba-inc.com

## Abstract

Recent studies revealed that using third-party models may lead to backdoor threats, where adversaries can maliciously manipulate model predictions based on backdoors implanted during model training. Arguably, backdoor trigger inversion (BTI), which generates trigger patterns of given benign samples for a backdoored model, is the most critical module for backdoor defenses used in these scenarios. With BTI, defenders can remove backdoors by fine-tuning based on generated poisoned samples with ground-truth labels or deactivate backdoors by removing trigger patterns during the inference process. However, we find that existing BTI methods suffer from relatively poor performance, *i.e.*, their generated triggers are significantly different from the ones used by the adversaries even in the feature space. We argue that it is mostly because existing methods require to 'extract' backdoor features at first, while this task is very difficult since defenders have no information (*e.g.*, trigger pattern or target label) about poisoned samples. In this paper, we explore BTI from another perspective where we decouple benign features instead of decoupling backdoor features directly. Specifically, our method consists of two main steps, including **(1)** decoupling benign features and **(2)** trigger inversion by minimizing the differences between benign samples and their generated poisoned version in decoupled benign features while maximizing the differences in remaining backdoor features. In particular, our method is more efficient since it doesn't need to 'scan' all classes to speculate the target label, as required by existing BTI. We also exploit our BTI module to further design backdoor-removal and pre-processing-based defenses. Extensive experiments on benchmark datasets demonstrate that our defenses can reach state-of-the-art performances. Our codes are available at `https://github.com/xuxiong0214/BTIDBF`.

## 1 Introduction

Deep neural networks (DNNs) play an important role in many mission-critical applications (Li et al., 2014; He et al., 2022; Bai et al., 2022; Sun et al., 2023; Li et al., 2023a; Yao et al., 2024). Currently, training a well-performed model is generally consuming or even expensive, requiring many high-quality samples and computational resources. Accordingly, many developers will directly adopt open-sourced DNNs from third-party model zoos (*e.g.*, Hugging Face) for their further development.

However, recent studies (Gu et al., 2019; Li et al., 2022c; Dong et al., 2023) revealed that using third-party models may lead to backdoor threats, where adversaries can maliciously manipulate the prediction of backdoored DNNs with pre-defined trigger patterns, based on backdoors implanted during training. In particular, attacked DNNs behave normally on benign samples. As such, these attacks are very stealthy since model users can hardly notice them based on their local benign samples.

Currently, there are also some defenses (Huang et al., 2022; Wang et al., 2022b; Guo et al., 2023a; Wang et al., 2023; Gao et al., 2023a) designed to reduce backdoor threats. Arguably, backdoor trigger

---

[*]The first three authors contributed equally to this work. † indicates corresponding authors: Yiming Li (li-ym@zju.edu.cn) and Zhan Qin (qinzhan@zju.edu.cn).

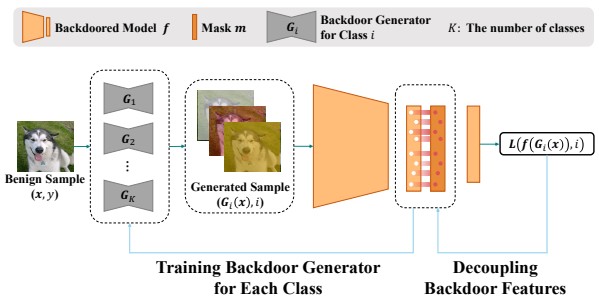

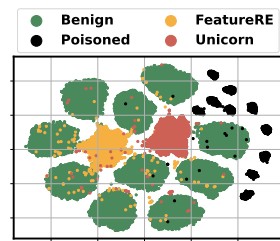

Figure 1: The main pipeline of existing backdoor trigger inversion (BTI). In general, they need to train a generator for each class ($K$ in total) at first since defenders have no prior knowledge of backdoor attacks (*i.e.*, trigger patterns and the target label). After that, they decouple backdoor features based on trained generators.

Figure 2: The t-SNE visualization of different samples involved in existing backdoor trigger inversion (BTI) for IAD. The recovered poisoned samples generated by existing BTI methods (*i.e.*, FeatureRE and Unicorn) are significantly different from ground-truth poisoned samples.

inversion (BTI) is the most critical module for backdoor defenses for the scenarios of using third-party models. In general, BTI aims to recover trigger patterns of given benign samples that can activate the backdoor in attacked models. With BTI, defenders can 'unlearn' and fix hidden backdoors via generated poisoned images (*i.e.*, benign samples containing their corresponding trigger patterns) with the ground-truth instead of target labels; Defenders can also pre-process suspicious samples by removing trigger patterns before feeding them into model prediction to prevent backdoor activation.

However, we find that existing BTI methods suffer from two main problem, including **(1)** low inversion efficiency and **(2)** low similarity between inversed and ground-truth triggers even in the feature space. We argue that these limitations are all because they need to approximate and decouple backdoor features at first to separate benign and backdoor features, as required by BTI. Specifically, these methods need to 'scan' all potential classes to speculate the target label since defenders have no prior knowledge about attacks and poisoned samples. These processes are time-consuming since each scan requires iteratively solving a particular optimization problem (as shown in Figure 1); Defenders also need to assign a particular poisoning form (*e.g.*, $x' = m \cdot x + (1 - m) \cdot t$) to approximate backdoor features, no matter in the input space (Wang et al., 2019) or the hidden feature space (Wang et al., 2022b; 2023). However, the approximation could be inaccurate in many cases, leading to low similarity between inversed and ground-truth poisoned samples (as shown in Figure 2).

In this paper, we explore the feature decoupling of BTI from another perspective. Instead of directly approximating backdoor features, we decouple benign features to separate backdoor ones, inspired by (Qi et al., 2023b) to some extent. This approach is motivated by defenders having local benign samples directly related to benign features. In general, our method consists of two main steps. In the first step, we decouple benign features by optimizing the objective that the suspicious model can make correct predictions on benign samples via only benign features, whereas using the remaining ones will lead to wrong predictions. In the second step, we train a *backdoor generator* by minimizing the differences between benign samples and their generated poisoned version in decoupled benign features while maximizing the differences in remaining poisoned features. After that, we also exploit our BTI module to further design backdoor-removal and pre-processing-based defenses. Specifically, we fine-tune the attacked model with generated poisoned images whose label is marked as their ground-truth label instead of the target one to 'unlearn' and remove model backdoors; We train a *purification generator* to approximate the inverse function of backdoor generator, based on which to pre-process suspicious samples before feeding them into model prediction to deactivate hidden backdoors. In particular, we design an enhancement method for them by repeatedly updating their generators based on the results of their target objects (*i.e.*, purified model and samples).

In conclusion, the main contributions of this paper are four-fold. **(1)** We reveal the low efficiency and low similarity nature of existing backdoor trigger inversion (BTI) methods and their main reasons. **(2)** Based on our analyses, we propose the first BTI that decouples benign features instead of backdoor features in the first step. Accordingly, our BTI method is fundamentally more efficient and reliable. **(3)** We also exploit our BTI module to further design simple yet effective pre-processing-based defense and adopt it for unlearning backdoor. **(4)** We conduct comprehensive experiments on benchmark datasets to verify the effectiveness of our methods and their resistance to potential adaptive attacks.

## 2 RELATED WORK

### 2.1 BACKDOOR ATTACKS

Backdoor attacks are emerging yet critical training-phase security threats when the model training includes third-party resources (Li et al., 2022b; Lan et al., 2024; Liang et al., 2024). In general, the attacked models behave normally on benign testing samples; However, adversaries can maliciously manipulate their predictions whenever testing samples contain adversary-specified trigger patterns. The latent connection between *trigger patterns* and malicious predictions is called *backdoor*. In general, existing attacks can be divided into two main categories, as follows.

**Backdoor Attacks with Poisoned Labels.** These attacks are currently the most classical and effective methods, where the assigned target labels of poisoned samples differ from their ground-truth label. BadNets (Gu et al., 2019) is the first and the most representative backdoor attack with poisoned labels, where almost all follow-up attacks were designed based on its paradigm. Specifically, BadNets randomly selected a few benign samples from the original dataset and turned them into *poisoned samples* by stamping the *backdoor trigger* onto the (benign) image and changing their label with an attacker-specified *target label*. These generated poisoned samples associated with remaining benign ones will be released to victim users as the *poisoned dataset*. Besides, it also discussed all-to-all attack mode whose target label is related to the ground-truth label of poisoned samples. The follow-up attacks mainly focused on how to design trigger patterns and escape existing backdoor defenses. For example, (Chen et al., 2017) introduced trigger transparency to BadNets to ensure attack stealthiness; BppAttack (Wang et al., 2022c) exploited image quantization and dithering to design more stealthy and effective trigger patterns; WaNet (Nguyen & Tran, 2021) is the most effective sample-specific backdoor attack whose trigger patterns are sample-specific instead of sample-agnostic, although this concept was first introduced in IAD (Nguyen & Tran, 2020) and ISSBA (Li et al., 2021d). As such, it can circumvent many existing defenses; Most recently, (Qi et al., 2023a) also discussed how to circumvent existing backdoor defenses by breaking their latent separability assumption.

**Backdoor Attacks with Clean Labels.** (Turner et al., 2019) revealed that attacks with poisoned labels are fundamentally not stealthy to some extent, although their trigger patterns can be invisible. Specifically, dataset users can easily break these attacks by finding and removing samples whose labels are mismatched. To address this problem, they proposed to poison samples only from the target class. In particular, they conducted adversarial attacks or GAN modification before adding trigger patterns to reduce the effects of 'robust features' that could hinder learning trigger patterns. After that, (Zhao et al., 2020) exploited universal adversarial perturbations (Moosavi-Dezfooli et al., 2017) as trigger patterns to further improve attack effectiveness; Most recently, (Gao et al., 2023b) proposed to modify 'hard' instead of random samples for designing better clean-label attacks. Both (Zhao et al., 2020) and (Gao et al., 2023b) were designed based on the paradigm of (Turner et al., 2019).

Recently, a few works also exploited backdoor attacks for positive purposes (Lin et al., 2021; Li et al., 2022a; 2023b; Guo et al., 2023b; Tang et al., 2023a; Ya et al., 2024), which are out of our scope.

### 2.2 BACKDOOR DEFENSES

Currently, there are also some defenses designed to reduce backdoor threats. In general, existing backdoor defenses consists of five main categories, including **(1)** input-level backdoor detection (Zeng et al., 2021a; Huang et al., 2023; Guo et al., 2023a), **(2)** poison suppression (Li et al., 2021a; Huang et al., 2022; Tang et al., 2023b), **(3)** backdoor-removal defenses (Li et al., 2021b; Zeng et al., 2022; Li et al., 2024), **(4)** pre-processing-based defenses (Liu et al., 2017; Doan et al., 2020; Li et al., 2021c), and **(5)** model-level backdoor detection (Guo et al., 2022; Xiang et al., 2023; Wang et al., 2024a). In this paper, we focus only on backdoor-removal and pre-processing-based defenses since we target the scenarios of using third-party model and detection-based methods cannot directly purified suspicious objects (*i.e.*, models and samples).

**Backdoor-Removal Defenses.** These defenses directly removed backdoors contained in the suspicious DNNs. For example, (Li et al., 2021b) exploited knowledge distillation with a purified teacher model to guide the fine-tuning of the attacked student model; (Zeng et al., 2022) adopted the implicit gradients to re-train attacked models; (Chai & Chen, 2022) masked model weights that are sensitive to the trigger; (Wang et al., 2022b) decoupled the backdoor features and flipped them for defense.

**Pre-Processing-based Defenses.** These defenses modified input testing samples before feeding them into the suspicious model to destroy potential trigger patterns. Accordingly, the attacked model can still generate correct predictions on poisoned images since modified trigger patterns can no longer activate its backdoors. (Liu et al., 2017) proposed the first pre-processing-based defenses where they adopted pre-trained auto-encoder for modification. After that, (Li et al., 2021c) and (Zeng et al., 2021b) exploited other classical transformations (*e.g.*, spatial transformations) for pre-processing.

**Backdoor Trigger Inversion.** As described in the introduction, backdoor trigger inversion (BTI) is a critical module for many backdoor defenses. In general, we can separate existing BTI methods into two main categories based on the inversion spaces (*i.e.*, pixel, and feature space). Neural cleanse (Wang et al., 2019) is the first BTI. It generated potential trigger patterns (and their masks) via universal adversarial perturbations in the pixel space. After that, many subsequent methods were designed based on it. For example, (Tao et al., 2022) developed a new optimization method without using a mask; (Hu et al., 2022) reduced training iterations by encouraging a diverse set of trigger candidates; (Shen et al., 2021) uses K-arm optimization to reduce computational resources. Currently, the most advanced BTI methods were developed in the feature space (Wang et al., 2022b; 2023). Besides, two recent research (Zheng et al., 2023; Feng et al., 2023) discussed how to conduct BTI in the feature space under self-supervised learning, inspired by neural cleanse (Wang et al., 2019).

## 3 METHODOLOGY

### 3.1 PRELIMINARIES

**The Main Pipeline of (Poisoning-based) Backdoor Attacks.** Let $\mathcal{D} = \{(\boldsymbol{x}_i, y_i)\}_{i=1}^N$ denotes the benign training set containing $N$ samples, where $\boldsymbol{x}_i \in \mathcal{X}$ is $i$-th image, $y_i \in \mathcal{Y} = \{1, \ldots, K\}$ is its label, and $K$ is the number of classes. The adversaries will generate a poisoned dataset $\mathcal{D}_p$, based on which to train the attacked model either with standard loss or adversary-specified one. Specifically, $\mathcal{D}_p$ consists of two main parts, including the modified version of a selected subset (*i.e.*, $\mathcal{D}_s$) of $\mathcal{D}$ and a benign subset $\mathcal{D}_b$, *i.e.*, $\mathcal{D}_p = \mathcal{D}_m \cup \mathcal{D}_b$, where $\mathcal{D}_b \subset \mathcal{D}$, $\mathcal{D}_m = \{(\boldsymbol{x}', y') | \boldsymbol{x}' = G_X(\boldsymbol{x}), y' = G_Y(y), (\boldsymbol{x}, y) \in \mathcal{D}_s\}$, $\gamma \triangleq \frac{|\mathcal{D}_s|}{|\mathcal{D}|}$ is the *poisoning rate*, and $G_X \& G_Y$ are adversary-specified *poisoned image generator* and *poisoned label generator*, respectively. For example, $G_X(\boldsymbol{x}) = (\boldsymbol{1} - \boldsymbol{\alpha}) \odot \boldsymbol{x} + \boldsymbol{\alpha} \odot \boldsymbol{t}$, where $\boldsymbol{\alpha} \in \{0, 1\}^{C \times W \times H}$, $\boldsymbol{t} \in \mathcal{X}$ is the trigger pattern, and $\odot$ is the element-wise product in BadNets (Gu et al., 2019); $G_Y(y) = y_t$ where the *target label* $y_t \in \mathcal{Y}$ in all-to-one attacks, while $G_Y(y) = (y + 1) \mod K$ in all-to-all attacks. In the inference process of the backdoored model, given an 'unseen' image $\hat{\boldsymbol{x}}$ with ground-truth label $\hat{y}$, the model will predict $\hat{\boldsymbol{x}}$ as $\hat{y}$ while predicting its poisoned version $G_X(\hat{\boldsymbol{x}})$ as $G_Y(\hat{y})$.

**Threat Model.** In this paper, we focus on backdoor defenses via backdoor trigger inversion when using third-party pre-trained models. We assume the defenders have full access to the suspicious model and a few local benign samples, whereas they have no information about the attack.

**Defender's Goals.** In general, defenders have two main goals for the module of backdoor trigger inversion (BTI), including reliability and efficiency. Reliability requires that the generated poisoned samples are similar to the ground-truth ones in the feature space, while efficiency hopes the BTI is fast. For backdoor-removal and pre-processing-based defenses built via our BTI, we expect the model to finally predict benign and poisoned testing samples to their ground-truth class (*i.e.*, with high benign accuracy and low attack success rate).

### 3.2 BACKDOOR TRIGGER INVERSION VIA DECOUPLING BENIGN FEATURES (BTI-DBF)

Following the most classical setting in existing defenses (Huang et al., 2022; Wang et al., 2022b; 2023), we treat a model $f$ as having two disjoint parts, including fully-connected layers $S_b$ and its previous (convolutional) layers $S_a$, *i.e.*, $f(\boldsymbol{x}) \triangleq S_b \circ S_a(\boldsymbol{x})$. $S_a$ is the mapping from the input feature space to the feature space while $S_b$ denotes the one from the feature space to the output space.

In general, our BTI-DBF method consists of two main steps, **(1)** decoupling benign features and **(2)** trigger inversion by minimizing the differences between benign samples and their generated poisoned version in decoupled benign features while maximizing the differences in remaining backdoor features. The main pipeline of our method is shown in Figure 3. Please find more technical details of our BTI-DBF module and defenses based on it in the following parts.

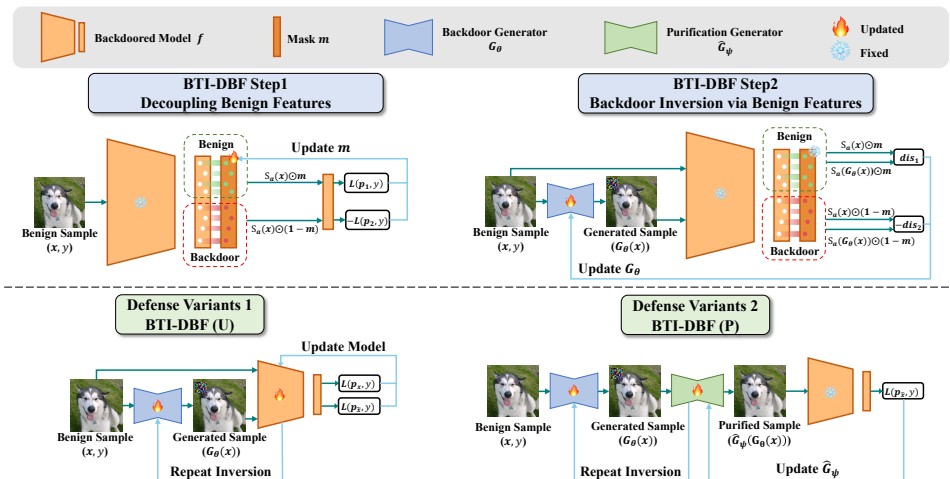

Figure 3: The main pipeline of our backdoor trigger inversion via decoupling benign features (BTI-DBF), as well as our backdoor-removal (dubbed 'BTI-DBF (U)') and pre-processing-based (dubbed 'BTI-DBF (P)') defenses designed based on BTI-DBF. In general, our BTI-DBF has two main steps, including **(1)** decoupling benign features with benign samples in the feature space generated by convolutional layers $S_a$ and **(2)** trigger inversion by minimizing the differences between benign samples and their generated poisoned version in decoupled benign features while maximizing the differences in remaining backdoor features; For BTI-DBF (U), we fine-tune the attacked model with poisoned images generated by our BTI-DBF whose labels are marked as their ground-truth labels to 'unlearn' backdoors; For BTI-DBF (P), we train a purification generator to approximate the inverse function of BTI-DBF's backdoor generator. The purification generator will pre-process all testing samples before feeding them into DNNs for predictions.

### 3.2.1 DECOUPLING BENIGN FEATURES

Intuitively, a suspicious model should make correct predictions on benign samples using only benign features, whereas using the remaining ones will lead to wrong predictions.

Specifically, let $m$ be the mask having the same dimension as that of all features $S_a(\cdot)$, whose elements are all in the range of $[0, 1]$. We use the mask $m$ to select benign features (in a soft way). The closer the element value is to 1, the more likely the corresponding feature is benign. Given a few local benign samples (with correct annotation), *i.e.*, $\mathcal{D}_l = \{(\boldsymbol{x}_i, y_i)\}_{i=1}^M$, the optimization process of our feature decoupling can be formulated as follows:

$$\underset{\boldsymbol{m}}{\operatorname{argmin}} \sum_{(\boldsymbol{x},y)\in\mathcal{D}_l} \left[ \mathcal{L}(S_b \circ (S_a(\boldsymbol{x}) \odot \boldsymbol{m}), y) - \mathcal{L}(S_b \circ (S_a(\boldsymbol{x}) \odot (\boldsymbol{1} - \boldsymbol{m})), y) \right], \qquad (1)$$

where $\mathcal{L}$ is the loss function (*e.g.*, cross-entropy) and $\odot$ is the element-wise product.

Arguably, our method is more reliable than existing BTI methods since we only rely on benign samples without needing to assign a particular poisoning form for approximation. Besides, our approach can automatically adjust the percentage of identified benign features without manually choosing the ratio. In particular, our method is highly efficient since we don't need to 'scan' all classes (as required by existing BTI methods) to determine potential target labels.

### 3.2.2 BACKDOOR TRIGGER INVERSION VIA DECOUPLED FEATURES

Once we obtain the feature mask $m$ via Eq.(1), we can train a backdoor generator $G_{\boldsymbol{\theta}} : \mathcal{X} \to \mathcal{X}$ to generate the poisoned version of any benign samples. In general, the benign samples and their poisoned version should have similar values in benign features but inconsistent values in the remaining backdoor features. Accordingly, we can formulate this optimization problem as follows:

$$\min_{\boldsymbol{\theta}} \sum_{(\boldsymbol{x},y)\in\mathcal{D}_l} \left( \|(S_a(\boldsymbol{x}) - S_a(G_{\boldsymbol{\theta}}(\boldsymbol{x}))) \odot \boldsymbol{m}\| - \|(S_a(\boldsymbol{x}) - S_a(G_{\boldsymbol{\theta}}(\boldsymbol{x}))) \odot (\boldsymbol{1} - \boldsymbol{m})\| \right),$$
$$\text{s.t.} \ \ \|\boldsymbol{x} - G_{\boldsymbol{\theta}}(\boldsymbol{x})\| \leq \tau, \ \forall (\boldsymbol{x}, y) \in \mathcal{D}_l, \qquad (2)$$

where $\tau > 0$ is a hyper-parameter and $\|\cdot\|$ is a distance metric (*e.g.*, $\ell_2$-norm).

Notice that the backdoor generator $G_{\boldsymbol{\theta}}$ generates poisoned samples instead of trigger patterns for simplicity. We can easily obtain the trigger pattern of image $\boldsymbol{x}$ by $G_{\boldsymbol{\theta}}(\boldsymbol{x}) - \boldsymbol{x}$.

## 3.3 BACKDOOR DEFENSES VIA OUR BTI-DBF

In this section, we discuss how to further design backdoor-removal and pre-processing-based defenses based on our BTI-DBF module. The input-level backdoor detection derived directly by our pre-processing-based defense is demonstrated in Appendix A

### 3.3.1 BACKDOOR-REMOVAL DEFENSE VIA BTI-DBF

Given the trained backdoor generator $G_{\boldsymbol{\theta}}$ of BTI-DBF, we can fine-tune the attacked model $f_{\boldsymbol{w}}$ with generated poisoned images whose labels are marked as their ground-truth labels instead of target ones to 'unlearn' and remove model backdoors. Specifically, we require that both benign images and their generated poisoned versions can be correctly predicted as their ground-truth label while each pair lies closely in the feature space, as follows:

$$\min_{\boldsymbol{w}} \sum_{(\boldsymbol{x},y)\in\mathcal{D}_l} \mathcal{L}(f_{\boldsymbol{w}}(\boldsymbol{x}), y) + \mathcal{L}(f_{\boldsymbol{w}}(G_{\boldsymbol{\theta}}(\boldsymbol{x})), y) + \|S_a(\boldsymbol{x}) - S_a(G_{\boldsymbol{\theta}}(\boldsymbol{x}))\|. \tag{3}$$

In particular, as an enhancement, we can conduct the unlearning process and update our backdoor generator alternately to further improve its performance. More details are in Appendix B.1.

### 3.3.2 PRE-PROCESSING-BASED DEFENSE VIA BTI-DBF

To purify the poisoned samples, we need to train a purification generator $\hat{G}_{\boldsymbol{\psi}} : \mathcal{X} \to \mathcal{X}$. In general, given the backdoor generator $G_{\boldsymbol{\theta}}$, we can approximate its inverse function to obtain the purification generator. Specifically, we need to ensure that **(1)** the purification can remove backdoor triggers while **(2)** it does not influence benign samples. In this paper, we design a purification loss $\mathcal{L}_p$ and a benign loss $\mathcal{L}_b$ to approach them, respectively. The training of $\hat{G}_{\boldsymbol{\psi}}$ can be denoted as follows:

$$\min_{\boldsymbol{\psi}} \sum_{(\boldsymbol{x},y)\in\mathcal{D}_l} \mathcal{L}_p + \mathcal{L}_b, \tag{4}$$

where

$$\mathcal{L}_p = \mathcal{L}(f(\hat{G}_{\boldsymbol{\psi}}(G_{\boldsymbol{\theta}}(\boldsymbol{x}))), y) + \left\|S_a(\boldsymbol{x}) - S_a(\hat{G}_{\boldsymbol{\psi}}(G_{\boldsymbol{\theta}}(\boldsymbol{x})))\right\| + \left\|\boldsymbol{x} - \hat{G}_{\boldsymbol{\psi}}(G_{\boldsymbol{\theta}}(\boldsymbol{x}))\right\|, \tag{5}$$

$$\mathcal{L}_b = \mathcal{L}(f(\hat{G}_{\boldsymbol{\psi}}(\boldsymbol{x})), y) + \left\|S_a(\boldsymbol{x}) - S_a(\hat{G}_{\boldsymbol{\psi}}(\boldsymbol{x}))\right\| + \left\|\boldsymbol{x} - \hat{G}_{\boldsymbol{\psi}}(\boldsymbol{x})\right\|. \tag{6}$$

Similarly, we can also alternately update our purification generator and backdoor generator to further improve the defense performance. Please find more details in Appendix B.2.

## 4 EXPERIMENTS

### 4.1 MAIN SETTINGS

**Datasets and DNNs.** We conduct experiments on three benchmark datasets, including CIFAR-10 (Krizhevsky, 2009), GTSRB (Houben et al., 2013), and (a subset of) ImageNet (Deng et al., 2009). The ImageNet subset contains 100 classes. We evaluate our methods with ResNet-18 (He et al., 2016) on all datasets. We exploit U-Net (Ronneberger et al., 2015) as the structure of all generators. We randomly select 5% benign training samples as the local dataset for all defenses. Please refer to Appendix C.1 for more dataset details and Appendix D for results with more DNNs.

**Evaluation Metrics.** We adopt the distance between recovered poisoned samples and their ground-truth ones in the feature space to evaluate the reliability of BTI since existing backdoor attacks exhibit generalization in the input space (Qiao et al., 2019). Besides, we adopt the training time to assess

Table 1: The reliability evaluation of BTI methods on CIFAR-10. We adopt feature distance (FD) between recovered poisoned samples and their ground-truth ones and detection success rate (DSR, %) about whether BTI can correctly identify the target class for measurement. Among all methods, the best results are marked in boldface while failed cases (*i.e.*, DSR < 50%) are marked in red.

| BTI→ Attack↓, Metric→ | NC | | Pixel | | THTP | | FeatureRE | | Unicorn | | BTI-DBF | |
|---|---|---|---|---|---|---|---|---|---|---|---|---|
| | FD | DSR | FD | DSR | FD | DSR | FD | DSR | FD | DSR | FD | DSR |
| BadNets | 8.20 | **100** | 71.12 | **100** | 24.51 | **100** | 22.52 | 82 | 18.34 | **100** | **7.81** | 100 |
| Blended | 6.49 | **100** | 3.24 | **100** | 5.39 | 96 | 33.10 | 92 | 2.19 | 96 | **1.79** | 100 |
| WaNet | 0.41 | 16 | 0.36 | 16 | 0.31 | 28 | 0.85 | 18 | 0.50 | 86 | **0.15** | 100 |
| IAD | 0.09 | 18 | 0.10 | 16 | 0.09 | 24 | 0.13 | 22 | 0.09 | 18 | **0.03** | 100 |
| LC | **0.09** | 100 | 0.14 | 100 | 0.11 | 94 | 0.24 | 88 | 0.37 | 100 | 0.09 | 100 |
| BppAttack | 11.75 | 16 | 12.31 | 22 | 12.24 | 18 | 11.79 | 16 | 12.38 | 34 | **6.38** | 98 |

their efficiency. Following the most classical setting in backdoor-related works (Li et al., 2022b), we adopt the benign accuracy (BA) and attack success rate (ASR) to evaluate all defenses. The higher the BA, the lower the ASR, the better the defense.

**Baseline Selection for BTI.** We compare our BTI-DBF with five representative and advanced BTI methods, including **(1)** Neural Cleanse (dubbed 'NC') (Wang et al., 2019), **(2)** THTP (Hu et al., 2022), **(3)** Pixel (Tao et al., 2022), **(4)** FeatureRE (Wang et al., 2022b), and **(5)** Unicorn (Wang et al., 2023). The first three methods were designed on the input space, while the last two were designed on the feature space. Please find their detailed settings in Appendix C.2.

**Baseline Selection for Backdoor Defenses.** We compare our backdoor-removal defense via BTI-DBF (dubbed 'BTI-DBF (U)') with four representative and advanced backdoor-removal defenses, including **(1)** NAD (Li et al., 2021b), **(2)** I-BAU (Zeng et al., 2022), **(3)** AWM (Chai & Chen, 2022), and **(4)** FeatureRE (Wang et al., 2022b); We compare our pre-processing-based defense via BTI-DBF (dubbed 'BTI-DBF (P)') with two most representative methods, including Februus (Doan et al., 2020) and ShrinkPad (Li et al., 2021c). Please find their detailed settings in Appendix C.3.

**The Selection of Evaluated Backdoor Attacks.** We exploit six representative and advanced backdoor attacks, including **(1)** BadNets (Gu et al., 2019), **(2)** attack with blended strategy (dubbed 'Blended'), **(3)** label-consistent attack (dubbed 'LC') (Turner et al., 2019), **(4)** IAD (Nguyen & Tran, 2020), **(5)** WaNet (Nguyen & Tran, 2021), and **(6)** BppAttack (Wang et al., 2022c), to comprehensively evaluate the performance of different defenses. Please refer to Appendix C.4 for their detailed settings.

## 4.2 THE PERFORMANCE OF BACKDOOR TRIGGER INVERSION

In this section, we only provide the results on CIFAR-10 due to the space limitation. Please refer to Appendix E for more results on GTSRB and ImageNet.

**Main Results.** As shown in Table 1, our method can obtain more reliable triggers with lower distance between generated and ground-truth poisoned samples in the feature space, compared to existing BTI methods. For example, under the BadNets attack, the distance of our BTI-DBF is nearly 10 times smaller than that of Pixel. In particular, as shown in Figure 4, our method is significantly faster than all baseline methods. For example, our BTI-DBF needs only 60 seconds for training, which is more than 20 times faster than Unicorn. It is still more than 3 times faster than the most efficient BTI baseline (*i.e.*, Pixel). This efficiency advantage is even more pronounced in datasets with more classes (*e.g.*, GTSRB and ImageNet).

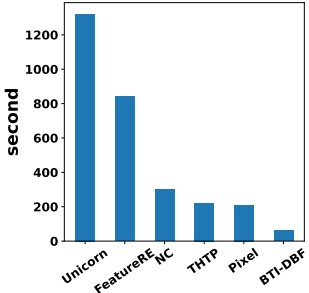

Figure 4: Training costs of BTI.

**A Closer Look to the Reliability of Our BTI-DBF.** To further explore why our method is highly reliable, we examine whether it can successfully find the target label for (all-to-one) backdoor attacks. Specifically, we train 10 models with different target labels and repeat all BTI methods for each model 5 times. We feed generated poisoned samples of each BTI method to backdoored DNNs and treat the predicted label having the highest frequency as the detected target label. We calculate the detection success rate (DSR) over all 50 trials for all methods at the end. As shown in Table 1, our method has 100% DSR in almost all cases. In contrast, all baseline methods fail (with DSR < 50%) in some cases. An interesting phenomenon is that there are methods with small distances in the feature samples (*e.g.*, NC and FeatureRE under WaNet and IAD) but they can not find the target

Table 2: The performance (%) of backdoor-removal defenses on CIFAR-10, GTSRB, and ImageNet datasets. We mark all failed cases (*i.e.*, BA drop or ASR > 10%) in red.

| Dataset↓ | Defense→ Attack↓ | No Defense | | NAD | | I-BAU | | AWM | | FeatureRE | | BTI-DBF (U) | |
|---|---|---|---|---|---|---|---|---|---|---|---|---|---|
| | | BA | ASR | BA | ASR | BA | ASR | BA | ASR | BA | ASR | BA | ASR |
| CIFAR-10 | BadNets | 92.82 | 99.88 | 90.32 | 2.98 | 90.67 | 1.33 | 90.93 | 21.92 | 91.53 | 35.42 | 92.00 | 1.36 |
| | Blended | 93.08 | 97.31 | 89.49 | 3.29 | 88.57 | 0.56 | 90.07 | 38.14 | 92.86 | 40.50 | 91.60 | 7.92 |
| | WaNet | 94.53 | 99.59 | 91.38 | 6.94 | 90.66 | 2.88 | 91.01 | 10.72 | 93.75 | 0.02 | 90.82 | 0.94 |
| | IAD | 94.07 | 99.41 | 91.34 | 17.45 | 91.11 | 9.63 | 88.76 | 14.71 | 93.23 | 0.39 | 91.91 | 1.22 |
| | LC | 94.65 | 88.83 | 91.77 | 12.65 | 93.12 | 1.60 | 92.07 | 12.61 | 94.54 | 10.49 | 90.48 | 4.51 |
| | BppAttack | 93.88 | 99.99 | 90.84 | 98.90 | 88.35 | 3.97 | 90.95 | 15.18 | 91.85 | 100 | 90.98 | 5.02 |
| GTSRB | BadNets | 97.14 | 100 | 95.41 | 3.35 | 95.70 | 0.00 | 96.80 | 0.43 | 97.09 | 100 | 95.35 | 0.04 |
| | Blended | 96.64 | 99.97 | 92.94 | 0.91 | 95.78 | 0.03 | 95.75 | 0.38 | 92.19 | 100 | 94.84 | 0.38 |
| | WaNet | 97.87 | 99.96 | 95.19 | 2.92 | 96.38 | 0.00 | 95.95 | 71.28 | 97.88 | 11.23 | 95.56 | 0.00 |
| | IAD | 97.22 | 99.81 | 94.39 | 0.38 | 95.55 | 0.05 | 94.83 | 9.67 | 97.23 | 0.00 | 93.64 | 0.00 |
| | BppAttack | 97.66 | 99.98 | 94.12 | 1.43 | 95.21 | 0.02 | 97.18 | 81.51 | 94.59 | 62.98 | 94.82 | 0.01 |
| ImageNet | BadNets | 73.23 | 99.98 | 70.25 | 13.50 | 71.59 | 98.91 | 71.80 | 48.02 | 72.42 | 48.82 | 70.49 | 7.62 |
| | Blended | 72.38 | 99.99 | 69.94 | 72.47 | 53.47 | 98.41 | 70.45 | 99.82 | 72.54 | 99.99 | 71.11 | 5.42 |
| | WaNet | 74.44 | 99.85 | 70.46 | 8.54 | 72.87 | 95.94 | 73.90 | 13.90 | 73.86 | 5.39 | 70.35 | 3.38 |
| | IAD | 73.76 | 99.76 | 70.85 | 35.58 | 54.61 | 15.22 | 67.56 | 47.36 | 72.66 | 27.48 | 69.93 | 5.52 |
| | BppAttack | 63.33 | 99.83 | 59.44 | 25.61 | 58.92 | 96.09 | 56.73 | 11.30 | 61.43 | 99.65 | 60.69 | 8.58 |

Table 3: The performance (%) of pre-processing-based defenses on CIFAR-10, GTSRB, and ImageNet datasets. We mark all failed cases (*i.e.*, BA drop or ASR > 10%) in red.

| Dataset | Attack | No Defense | | Februus | | ShrinkPad | | BTI-DBF (P) | |
|---|---|---|---|---|---|---|---|---|---|
| | | BA | ASR | BA | ASR | BA | ASR | BA | ASR |
| CIFAR-10 | BadNets | 92.82 | 99.88 | 90.14 | 2.22 | 84.21 | 2.04 | 90.28 | 1.23 |
| | Blended | 93.08 | 97.31 | 82.92 | 5.04 | 82.69 | 75.13 | 89.13 | 1.00 |
| | WaNet | 94.53 | 99.59 | 69.36 | 49.55 | 45.60 | 96.56 | 89.14 | 1.60 |
| | IAD | 94.07 | 99.41 | 66.45 | 32.40 | 88.14 | 35.83 | 90.21 | 3.73 |
| | LC | 94.65 | 88.83 | 71.51 | 16.95 | 88.37 | 2.13 | 90.02 | 1.11 |
| | BppAttack | 93.88 | 99.99 | 91.31 | 0.03 | 83.59 | 51.95 | 89.39 | 2.52 |
| GTSRB | BadNets | 97.14 | 100 | 65.27 | 0.04 | 96.38 | 0.00 | 93.30 | 1.10 |
| | Blended | 96.64 | 99.97 | 79.07 | 0.47 | 96.24 | 2.36 | 94.03 | 0.48 |
| | WaNet | 97.87 | 99.96 | 28.05 | 32.33 | 59.30 | 100 | 94.26 | 0.00 |
| | IAD | 97.22 | 99.81 | 30.28 | 36.89 | 96.98 | 39.55 | 93.95 | 0.00 |
| | BppAttack | 97.66 | 99.98 | 85.40 | 0.06 | 94.59 | 28.05 | 93.77 | 0.76 |
| ImageNet | BadNets | 73.23 | 99.98 | 34.03 | 2.83 | 70.66 | 11.13 | 68.63 | 7.90 |
| | Blended | 72.38 | 99.99 | 33.01 | 66.23 | 68.96 | 99.08 | 68.87 | 1.41 |
| | WaNet | 74.44 | 99.85 | 33.88 | 32.64 | 71.18 | 96.35 | 69.36 | 0.14 |
| | IAD | 73.76 | 99.76 | 33.25 | 54.70 | 70.32 | 96.78 | 69.52 | 9.40 |
| | BppAttack | 63.33 | 99.83 | 41.93 | 92.06 | 59.76 | 97.79 | 59.99 | 6.27 |

label correctly. It partly explain why defenses based on these BTI methods suffer from relatively low effectiveness. We will further explore its intrinsic mechanism in our future work.

## 4.3 THE PERFORMANCE OF BACKDOOR DEFENSES

**Results of Backdoor-removal Defenses.** As shown in Table 2, our BTI-DBF (U) can successfully remove model backdoors in all cases while preserving high benign accuracy. Specifically, the attack success rates of our method are lower than 10% and the drops of benign accuracy compared to the one with no defense are smaller than 5% in all cases. In contrast, all baseline defenses may fail (with ASR > 10%) even when they have significantly reduced benign accuracy. For example, the ASR of I-BAU is still higher than 95% even it has decreased BA for nearly 20% in defending against Blended on ImageNet. These results verify the effectiveness of our BTI-DBF (U) and our BTI-DBF module.

**Results of Pre-processing-based Defenses.** As shown in Table 3, our BTI-DBF (P) also performs best among all pre-processing-based defenses. Specifically, the ASRs of our method are lower than 10% (< 5% in most cases), while its BA drops are all smaller than 6%. In contrast, Februus and ShrinkPad fail in many cases, especially when trigger patterns are relatively large or sample-specific. This failure is caused by their non-essential assumptions of trigger patterns. These results verify the effectiveness of our BTI-DBF (P) and our BTI-DBF module again.

Table 4: The reliability of BTI-DBF without or with decoupling benign feature.

| | w/o DBF | | w/ DBF | |
|---|---|---|---|---|
| | FD | DSR | FD | DSR |
| BadNets | 8.33 | 68 | **7.81** | **100** |
| Blended | **1.53** | 42 | 1.79 | **100** |
| WaNet | 0.41 | 24 | **0.15** | **100** |
| IAD | 0.13 | 22 | **0.03** | **100** |
| LC | **0.09** | 58 | **0.09** | **100** |
| BppAttack | 10.34 | 44 | **6.38** | 98 |

Table 5: The performance (%) of designed defenses without or with iteration-based enhancement.

| | BTI-DBF (P) | | | | BTI-DBF (U) | | | |
|---|---|---|---|---|---|---|---|---|
| | w/o IE | | w/ IE | | w/o IE | | w/ IE | |
| | BA | ASR | BA | ASR | BA | ASR | BA | ASR |
| BadNets | 89.18 | 16.27 | **90.28** | 1.23 | 92.42 | 21.54 | 92.00 | **1.36** |
| Blended | 88.60 | 6.41 | **89.13** | 1.00 | 92.71 | 31.79 | 91.60 | 7.92 |
| WaNet | **89.97** | 6.82 | 89.14 | 1.60 | 92.52 | 10.51 | 90.82 | 0.94 |
| IAD | 89.65 | 21.18 | **90.21** | 3.37 | 92.04 | 6.84 | 91.91 | 1.22 |
| LC | **91.14** | 6.52 | 90.48 | 4.15 | 91.90 | 1.31 | 90.02 | 1.11 |
| BppAttack | 88.74 | 2.77 | **89.39** | 2.52 | 91.03 | 42.31 | 90.98 | 5.02 |

Table 6: The resistance to adaptive attacks on the CIFAR-10 dataset.

| | No Defense | | BTI-DBF (P) | | BTI-DBF (U) | |
|---|---|---|---|---|---|---|
| | BA | ASR | BA | ASR | BA | ASR |
| BadNets | 92.82 | 99.88 | 90.28 | 1.23 | 92.00 | 1.36 |
| Adap-Blended | 94.68 | 81.46 | 90.50 | 4.84 | 92.28 | 1.85 |
| Adaptive BadNets | 92.52 | 99.98 | 89.72 | 5.32 | 91.05 | 0.78 |

## 4.4 ABLATION STUDY

There are two important components in our methods, including **(1)** decoupling benign features in our BTI-DBF module and **(2)** iteration-based enhancement in designed backdoor defenses. In this section, we verify their effectiveness. Specifically, we conduce experiments on CIFAR-10. Unless otherwise specified, all settings are the same as those stated in Section 4.1.

**Effectiveness of Decoupling Benign Feature in BTI-DBF.** We hereby conduct our trigger inversion process on all features (with mask $m = 0$) instead of decoupled features via Eq. (2) and calculate the feature distance (FD) and detection success rate (DSR) for evaluation. As shown in Table 4, BTI-DBF is better than its variant without feature decoupling in almost all cases. In particular, in most cases, BTI-DBF without feature decoupling cannot correctly identify the target class (with DSR $< 50\%$). These results verify the effectiveness of this component.

**Effectiveness of Iteration-based Enhancement in Defenses.** Recall that we propose to alternately update our target object (*i.e.*, model or purification generator) as an iteration-based enhancement (IE) to further improve our defenses. As shown in Table 5, our IE can significantly ASR. These results verify the effectiveness of our iteration-based enhancement.

## 4.5 THE RESISTANCE TO POTENTIAL ADAPTIVE ATTACKS

In this section, we analyze whether adversaries can easily break our defenses if they know our method.

**Settings.** In general, our methods also implicitly rely on the latent separation assumption of backdoor attacks to some extent, since we try to decouple benign and backdoor features. We hereby adopt Adap-Blended (Qi et al., 2023a) as the adaptive attack since it can reduce latent separation between benign and poisoned samples. In particular, we also design an adaptive method (dubbed 'Adaptive BadNets') to directly target our BTI-DBF, where we require that using backdoor features can also correctly classify benign samples. Please find more details in Appendix I.

**Results.** As shown in Table 6, our methods are still highly effective under these attacks with high benign accuracy and low attack success rate, although their performance may have a few degrades. In other words, our methods are resistant to adaptive attacks.

## 5 CONCLUSION

In this paper, we proposed the first backdoor trigger inversion (BTI) that decouples benign features instead of backdoor features, which is effective and highly efficient. This method is motivated by our analyses of why existing BTI methods have low efficiency and similarity. We also designed simple yet effective backdoor-removal and pre-processing-based defenses based on it. Results on benchmark datasets verified the effectiveness of our methods and their resistance to potential adaptive attacks. We hope our method can provide a deeper understanding of backdoor trigger inversion to facilitate the design of more effective backdoor defenses and secure DNNs.

## ACKNOWLEDGEMENT

This work was supported by the National Key Research and Development Program of China under Grant 2021YFB3100300 and the National Natural Science Foundation of China under Grants U20A20178, 62072395, and 62206207.

## ETHICS STATEMENT

Backdoor attacks are serious threats when victim users exploit untrusted third-party resources for model training. In this paper, we propose a new framework of backdoor trigger inversion, based on which we design two simple yet effective backdoor defenses. Accordingly, this work has no ethical issue in general since our work is purely defensive and does not discover any new threats. However, we must emphasize that our methods are available in using suspicious third-party models only when defenders have (a few) local samples, whereas backdoor attacks could also happen in other scenarios (*e.g.*, using third-party samples). In these cases, people should only use trusted resources or other particular defenses. People should not be too optimistic about eliminating backdoor threats.

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

Table 7: The performance of our BTI-DBF (D) on the CIFAR-10 dataset.

| Metric↓, Attack→ | BadNets | Blended | WaNet | IAD | LC | BppAttack |
|---|---|---|---|---|---|---|
| Precision (%) | 93.28 | 92.66 | 91.50 | 92.51 | 92.01 | 93.49 |
| Recall (%) | 99.01 | 98.44 | 97.26 | 93.70 | 88.53 | 99.00 |
| F1-Score (%) | 96.06 | 95.47 | 94.29 | 93.10 | 90.24 | 96.16 |

---

**Algorithm 1** The algorithm of our BTI-DBF (U).

---

**Input:** The suspicious $f_{\boldsymbol{w}}$ with parameters $\boldsymbol{w}$, a small set of local benign samples $\mathcal{D}_l$, hyper-parameter $\tau$, step sizes of updates $\alpha_1$, $\alpha_2$, and $\alpha_3$, the number of iterations for decoupling benign features $I_1$, the number of iterations for training backdoor generator $I_2$, the number of iterations for unlearning backdoor $I_3$, and the numbers of alternating iterations $R_1$.

**Output:** Purified model $f_{\boldsymbol{w}}$.

1: **for** $r$ in $range(R_1)$ **do**
2:     Initial $\boldsymbol{m} \leftarrow \boldsymbol{0}$
3:     **for** $i$ in $range(I_1)$ **do**
4:         $\ell_{\boldsymbol{m}} \leftarrow \mathcal{L}(S_b \circ (S_a(\boldsymbol{x}) \odot \boldsymbol{m}), y) - \mathcal{L}(S_b \circ (S_a(\boldsymbol{x}) \odot (\boldsymbol{1} - \boldsymbol{m})), y)$
5:         Update $\boldsymbol{m} \leftarrow \boldsymbol{m} - \alpha_1 \cdot \nabla_{\boldsymbol{m}} \ell_{\boldsymbol{m}}$
6:     **for** $j$ in $range(I_2)$ **do**
7:         **if** $\|\boldsymbol{x} - G_{\boldsymbol{\theta}}(\boldsymbol{x})\| \leq \tau$ **then**
8:             $\ell_{\boldsymbol{\theta}} \leftarrow \|(S_a(\boldsymbol{x}) - S_a(G_{\boldsymbol{\theta}}(\boldsymbol{x}))) \odot \boldsymbol{m}\| - \|(S_a(\boldsymbol{x}) - S_a(G_{\boldsymbol{\theta}}(\boldsymbol{x}))) \odot (\boldsymbol{1} - \boldsymbol{m})\|$
9:         **else**
10:             $\ell_{\boldsymbol{\theta}} \leftarrow \|\boldsymbol{x} - G_{\boldsymbol{\theta}}(\boldsymbol{x})\|$
11:         Update $\boldsymbol{\theta} \leftarrow \boldsymbol{\theta} - \alpha_2 \cdot \nabla_{\boldsymbol{\theta}} \ell_{\boldsymbol{\theta}}$
12:     **for** $k$ in $range(I_3)$ **do**
13:         $\ell_{\boldsymbol{w}} \leftarrow \mathcal{L}(f_{\boldsymbol{w}}(\boldsymbol{x}), y) + \mathcal{L}(f_{\boldsymbol{w}}(G_{\boldsymbol{\theta}}(\boldsymbol{x})), y) + \|S_a(\boldsymbol{x}) - S_a(G_{\boldsymbol{\theta}}(\boldsymbol{x}))\|$
14:         Update $\boldsymbol{w} \leftarrow \boldsymbol{w} - \alpha_3 \cdot \nabla_{\boldsymbol{w}} \ell_{\boldsymbol{w}}$

---

## A   Design Input-Level Backdoor Detection based on BTI-DBF

In general, we can easily design an input-level backdoor detection (dubbed 'BTI-DBF (D)') based on the pre-processing-based defense. Specifically, for each suspicious testing sample, we feed it and its purified version generated by our BTI-DBF (P) and examine their predictions. If their predicted labels are consistent, we treat the suspicious sample as benign; otherwise, it is regarded as poisoned.

We adopt three classical metrics, including precision, recall, and F1-score, to evaluate the performance of our BTI-DBF (D) on the CIFAR-10 dataset. As shown in Table 7, our BTI-DBF (D) is highly effective in detecting all backdoor attacks due to the excellent performance of our BTI-DBF (P).

## B   More Technical Details of the Iteration-based Enhancement

In this section, we provide more technical details of the iteration-based enhancement for our designed defenses (*i.e.*, BTI-DBF (U) and BTI-DBF (P)).

### B.1   Iteration-based Enhancement for BTI-DBF (U)

As we mentioned in Section 3.3.1, we can conduct the unlearning process and update our backdoor generator alternately to improve BTI-DBF (U). Its technical details are shown in Algorithm 1.

Specifically, we conduct BTI-DBF in Line 2-11 and conduct backdoor unlearning in Line 12-14. In particular, we decouple features in Line 2-5 and learn our backdoor generator in Line 6-11.

**Remark 1.** *In each loop, we omit the partition of $\mathcal{D}_l$ with given batch size and the mini-batch SGD.*

**Remark 2.** *We repeat the training of backdoor generator and backdoor unlearning $R_1$ times, where $S_a$ and $S_b$ will be changed in each time due to the update of model parameters $\boldsymbol{w}$.*

---

**Algorithm 2** The algorithm of BTI-DBF (P).

---

**Input:** The suspicious model $f$, a small set of local benign samples $\mathcal{D}_l$, hyper-parameter $\tau$, step sizes of updates $\alpha_1$, $\alpha_2$, and $\alpha_3$, the number of iterations for decoupling benign features $I_1$, the number of iterations for training backdoor generator $I_2$, the number of iterations for training purification generator $I_3$, and the numbers of alternating iterations $R_1$.

**Output:** The purification generator $\hat{G}_{\boldsymbol{\psi}}$.

1: **for** $r$ in $range(R_1)$ **do**
2:      Initial $\boldsymbol{m} = \mathbf{0}$
3:      **for** $i$ in $range(I_1)$ **do**
4:          $\ell_{\boldsymbol{m}} \leftarrow \mathcal{L}(S_b \circ (S_a(\boldsymbol{x}) \odot \boldsymbol{m}), y) - \mathcal{L}(S_b \circ (S_a(\boldsymbol{x}) \odot (\mathbf{1} - \boldsymbol{m})), y)$
5:          Update $\boldsymbol{m} \leftarrow \boldsymbol{m} - \alpha_1 \cdot \nabla_{\boldsymbol{m}} \ell_{\boldsymbol{m}}$
6:      **for** $j$ in $range(I_2)$ **do**
7:          **if** $\|\boldsymbol{x} - G_{\boldsymbol{\theta}}(\boldsymbol{x})\| \leq \tau$ **then**
8:             **if** $r = 0$ **then**
9:                $\ell_1 \leftarrow \|(S_a(\boldsymbol{x}) - S_a(G_{\boldsymbol{\theta}}(\boldsymbol{x}))) \odot \boldsymbol{m}\|$
10:               $\ell_2 \leftarrow \|(S_a(\boldsymbol{x}) - S_a(G_{\boldsymbol{\theta}}(\boldsymbol{x}))) \odot (\mathbf{1} - \boldsymbol{m})\|$
11:             **else**
12:                $\ell_1 \leftarrow \left\|(S_a(\boldsymbol{x}) - S_a(\hat{G}_{\boldsymbol{\psi}}(G_{\boldsymbol{\theta}}(\boldsymbol{x})))) \odot \boldsymbol{m}\right\|$
13:                $\ell_2 \leftarrow \left\|(S_a(\boldsymbol{x}) - S_a(\hat{G}_{\boldsymbol{\psi}}(G_{\boldsymbol{\theta}}(\boldsymbol{x})))) \odot (\mathbf{1} - \boldsymbol{m})\right\|$
14:             $\ell_{\boldsymbol{\theta}} \leftarrow \ell_1 - \ell_2$
15:          **else**
16:             $\ell_{\boldsymbol{\theta}} \leftarrow \|\boldsymbol{x} - G_{\boldsymbol{\theta}}(\boldsymbol{x})\|$
17:          Update $\boldsymbol{\theta} \leftarrow \boldsymbol{\theta} - \alpha_2 \cdot \nabla_{\boldsymbol{\theta}} \ell_{\boldsymbol{\theta}}$
18:      **for** $k$ in $range(I_3)$ **do**
19:          $\ell_p \leftarrow \mathcal{L}(f(\hat{G}_{\boldsymbol{\psi}}(G_{\boldsymbol{\theta}}(\boldsymbol{x}))), y) + \left\|S_a(\boldsymbol{x}) - S_a(\hat{G}_{\boldsymbol{\psi}}(G_{\boldsymbol{\theta}}(\boldsymbol{x})))\right\| + \left\|\boldsymbol{x} - \hat{G}_{\boldsymbol{\psi}}(G_{\boldsymbol{\theta}}(\boldsymbol{x}))\right\|$
20:          $\ell_b \leftarrow \mathcal{L}(f(\hat{G}_{\boldsymbol{\psi}}(\boldsymbol{x})), y) + \left\|S_a(\boldsymbol{x}) - S_a(\hat{G}_{\boldsymbol{\psi}}(\boldsymbol{x}))\right\| + \left\|\boldsymbol{x} - \hat{G}_{\boldsymbol{\psi}}(\boldsymbol{x})\right\|$
21:          $\ell_{\boldsymbol{\psi}} = \ell_p + \ell_b$
22:          Update $\boldsymbol{\psi} \leftarrow \boldsymbol{\psi} - \alpha_3 \cdot \nabla_{\boldsymbol{\psi}} \ell_{\boldsymbol{\psi}}$

---

### B.2 ITERATION-BASED ENHANCEMENT FOR BTI-DBF (P)

As we mentioned in Section 3.3.2, we can also alternately update our purification generator and backdoor generator to improve BTI-DBF (P). Its technical details are shown in Algorithm 2.

Specifically, we conduct BTI-DBF in Line 3-17 and train purification generator in Line 18-22. In particular, we decouple features in Line 3-5 and learn our backdoor generator in Line 6-17.

**Remark 3.** *In each loop, we omit the partition of $\mathcal{D}_l$ with given batch size and the mini-batch SGD.*

## C MORE DETAILS OF MAIN SETTINGS

### C.1 DATASETS

**CIFAR-10.** The CIFAR-10 dataset (Krizhevsky, 2009) consists of 50,000 training samples and 10,000 testing samples. Each image belongs to one of ten classes and with size $3 \times 32 \times 32$.

**GTSRB.** German traffic sign recognition benchmark (GTSRB) (Houben et al., 2013) is a dataset used for traffic sign recognition, having 43 different classes. The number of training samples and testing samples are are 39,209 and 12,630, respectively. In this paper, we resize all images to the size of $3 \times 32 \times 32$. Please note that different classes may have different numbers of samples.

**ImageNet.** It is a large-scale image dataset for visual object recognition (Deng et al., 2009), containing over 14 million manually annotated images. In this paper, we select a subset from the original ImageNet dataset. Specifically, it contains 100 classes where each class has 500 training and 100 testing images with size $3 \times 224 \times 224$.

Table 8: The summary of adopted datasets.

| Dataset | Image Size | # Train Samples | # Test Samples | # Classes |
|---------|-----------|-----------------|----------------|-----------|
| CIFAR-10 | $3 \times 32 \times 32$ | 50,000 | 10,000 | 10 |
| GTSRB | $3 \times 32 \times 32$ | 39,209 | 12,630 | 43 |
| ImageNet | $3 \times 224 \times 224$ | 50,000 | 10,000 | 100 |

Their information is summarized in Table 8.

## C.2 SETTINGS FOR BTI METHODS

In this section, we describe the detailed settings used in our main experiments.

**NC.** We follow the default settings used in its original paper (Wang et al., 2019).

**Pixel.** We use the same method as NC to detect the target label, other settings are the same as those used in their original paper (Tao et al., 2022).

**THTP.** We follow the default settings used in its original paper (Hu et al., 2022).

**FeatureRE.** We follow the default settings used in its original paper (Wang et al., 2022b).

**Unicorn.** We follow the default settings used in its original paper (Wang et al., 2022b).

**BTI-DBF.** In our paper, we set the number of iterations for decoupling benign features $I_1 = 20$, the number of iterations for training backdoor generator $I_2 = 30$, the step size $\alpha_1 = 0.01$, $\alpha_2 = 0.01$, and hyper-parameter $\tau = 0.3$ in BTI-DBF.

## C.3 SETTINGS FOR BACKDOOR DEFENSES

**NAD.** We follow the default settings used in its original paper (Li et al., 2021b) on CIFAR-10 and GTSRB. We follow the settings used in (Huang et al., 2022) on ImageNet.

**I-BAU.** We follow the default settings used in its original paper (Zeng et al., 2022).

**AWM.** We follow the default settings in its original paper(Wang et al., 2022b).

**FeatureRE.** We follow the default settings in its original paper(Wang et al., 2022b).

**BTI-DBF (U).** We set the number of iterations for unlearning backdoor $I_3 = 20$, step size $\alpha_3 = 0.001$, and the number of alternating iterations $R_1 = 5$.

**Februus.** We follow the default settings used in its original paper (Li et al., 2021b) on CIFAR-10 and GTSRB. We use the pre-trained inpainting model, following the setting of (Yu et al., 2018).

**ShrinkPad.** We set the shrinking size as 2, other settings are the same as their original paper (Li et al., 2021c).

**BTI-DBF (P).** We set the number of iterations for training purification generator $I_3 = 30$, step size $\alpha_3 = 0.001$, and the number of alternating iterations $R_1 = 5$.

## C.4 SETTINGS FOR BACKDOOR ATTACKS

**BadNets.** In this paper, we use a $2 \times 2$ random square as the trigger pattern on CIFAR-10 and GTSRB and a $32 \times 32$ square on ImageNet. Other settings are the same as those used in (Gu et al., 2019).

**Blended.** In this paper, we set the trigger pattern as random Gaussian noise and set the blended ratio as 0.1 on all datasets. Other settings are the same as those used in (Chen et al., 2017).

**WaNet.** We follow the default settings used in its original paper (Nguyen & Tran, 2021).

**IAD.** We follow the default settings used in its original paper(Nguyen & Tran, 2020).

**LC.** We use the projected gradient descent (PGD) to make adversarial samples and set the maximum perturbation size $\epsilon = 8$. The trigger patterns are the same as those used in BadNets.

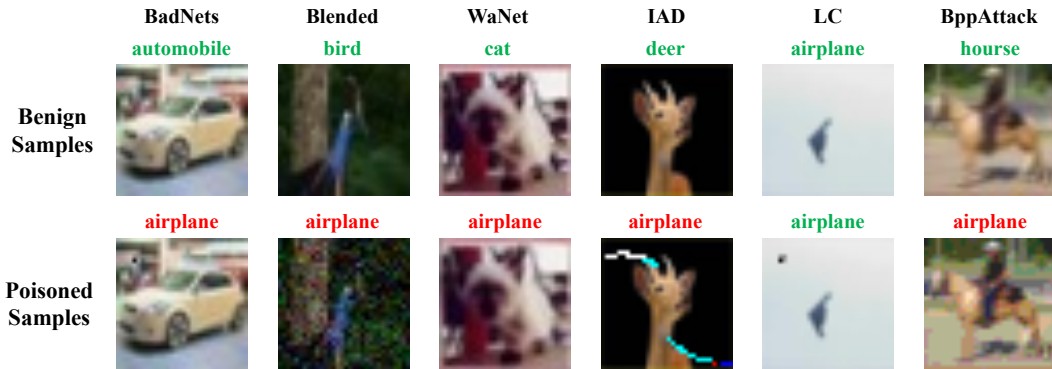

Figure 5: The example of poisoned samples generated by different backdoor attacks and their benign version on the CIFAR-10 dataset. In these cases, the target label is 'airplane'. In particular, we mark consistent labels in green and inconsistent ones in red.

**BppAttack.** We follow the default settings used in its original paper (Wang et al., 2022c).

The example of poisoned samples generated by different attacks on CIFAR-10 is shown in Figure 5.

### C.5 Setting for Model Training

We adopt the SGD with a momentum of 0.9 and a weight decay of $5 \times 10^{-4}$ as the optimizer to train all attacked DNNs. Specifically, we set the batch size as 128 on CIFAR-10 and GTSRB, while it is set to 32 on ImageNet. We set the initial learning rate as 0.1 and train all models 300 epochs. The learning rate will be multiplied by a factor of 0.1 at 100-th epoch.

### C.6 Setting for Computational Facilities

In this paper, we run all experiments on a single RTX 3090 Ti GPU with PyTorch.

## D Additional Results with More DNNs

In our main experiments, we evaluate all methods only with ResNet-18. In this section, we provide additional results with more representative DNNs.

Specifically, we conduct experiments on CIFAR-10 with three other DNNs, including VGG-16 (Simonyan & Zisserman, 2014), DenseNet-121 (Huang et al., 2017), and ViT (Dosovitskiy et al., 2020) (without pre-training). As shown in Table 9, our methods are still highly effective (*i.e.*, with high DSR&BA and low ASR) in all cases. These results confirm the generalizability of our methods.

## E Additional Results of Backdoor Trigger Inversion

In our main contents, we only provide the results on CIFAR-10. In this section, we provide the results on all three datasets (*i.e.*, CIFAR-10, GTSRB, ImageNet).

As shown in Table 10, our BTI-DBF can still reach the best performance on GTSRB and ImageNet. In particular, all baseline BTI methods fail in some cases, whereas our method is reliable in all cases.

## F The Effects of Key Hyper-parameters

In this section, we discuss the influence of key hyper-parameters involved in our methods. For simplicity, we conduct experiments on the CIFAR-10 dataset. Unless otherwise specified, all settings are the same as those stated in Section 4.1.

Table 9: The performance (%) of our methods under different model structures on CIFAR-10.

| Attack↓ | Defense→ Model↓, Metric→ | BTI-DBF DSR | No Defense BA | No Defense ASR | BTI-DBF (P) BA | BTI-DBF (P) ASR | BTI-DBF (U) BA | BTI-DBF (U) ASR |
|---|---|---|---|---|---|---|---|---|
| BadNets | VGG-16 | 96 | 91.72 | 99.73 | 87.06 | 3.31 | 90.50 | 4.50 |
| | DenseNet-121 | 90 | 91.98 | 99.93 | 88.17 | 7.80 | 90.45 | 2.44 |
| | ViT | 98 | 82.90 | 99.15 | 81.23 | 6.22 | 80.65 | 5.65 |
| Blended | VGG-16 | 100 | 90.70 | 100 | 86.96 | 4.52 | 90.36 | 4.18 |
| | DenseNet-121 | 94 | 92.34 | 100 | 89.52 | 5.21 | 90.63 | 5.48 |
| | ViT | 98 | 83.13 | 99.95 | 79.79 | 4.17 | 80.18 | 3.52 |
| WaNet | VGG-16 | 100 | 93.62 | 94.64 | 88.15 | 2.23 | 90.54 | 1.61 |
| | DenseNet-121 | 100 | 95.03 | 97.61 | 88.57 | 4.98 | 90.59 | 0.92 |
| | ViT | 88 | 84.24 | 94.65 | 81.32 | 6.57 | 80.30 | 2.05 |
| IAD | VGG-16 | 100 | 92.78 | 99.45 | 89.92 | 5.75 | 90.01 | 4.04 |
| | DenseNet-121 | 100 | 94.74 | 99.71 | 90.49 | 1.56 | 92.22 | 1.11 |
| | ViT | 100 | 80.74 | 92.76 | 77.31 | 5.19 | 78.94 | 1.07 |
| BppAttack | VGG-16 | 94 | 92.12 | 100 | 89.31 | 2.19 | 91.30 | 1.74 |
| | DenseNet-121 | 98 | 93.82 | 100 | 90.01 | 3.59 | 91.66 | 4.74 |
| | ViT | 100 | 87.47 | 99.75 | 84.25 | 4.53 | 83.55 | 5.11 |

Table 10: The reliability evaluation of BTI methods. We adopt feature distance (FD) between recovered poisoned samples and their ground-truth ones and detection success rate (DSR, %) about whether BTI can correctly identify the target class for measurement. Among all methods, the best results are marked in boldface while failed cases (*i.e.*, DSR < 50%) are marked in red.

| Dataset↓ | BTI→ Attack↓, Metric→ | NC FD | NC DSR | Pixel FD | Pixel DSR | THTP FD | THTP DSR | FeatureRE FD | FeatureRE DSR | Unicorn FD | Unicorn DSR | BTI-DBF FD | BTI-DBF DSR |
|---|---|---|---|---|---|---|---|---|---|---|---|---|---|
| CIFAR-10 | BadNets | 8.20 | **100** | 71.12 | **100** | 24.51 | **100** | 22.52 | 82 | 18.34 | **100** | **7.81** | 100 |
| | Blended | 6.49 | **100** | 3.24 | **100** | 5.39 | 96 | 33.10 | 92 | 2.19 | 96 | **1.79** | 100 |
| | WaNet | 0.41 | 16 | 0.36 | 16 | 0.31 | 28 | 0.85 | 18 | 0.50 | 86 | **0.15** | 100 |
| | IAD | 0.09 | 18 | 0.10 | 16 | 0.09 | 24 | 0.13 | 22 | 0.09 | 18 | **0.03** | 100 |
| | LC | **0.09** | 100 | 0.14 | 100 | 0.11 | 94 | 0.24 | 88 | 0.37 | 100 | 0.09 | 100 |
| | BppAttack | 11.75 | 16 | 12.31 | 22 | 12.24 | 18 | 11.79 | 16 | 12.38 | 34 | **6.38** | 98 |
| GTSRB | BadNets | 15.14 | **100** | 209.34 | **100** | 20.79 | **100** | 304.06 | **100** | 144.18 | **100** | **8.03** | 94 |
| | Blended | 6.98 | 98 | 4.32 | **100** | 6.02 | **100** | 22.59 | 92 | 15.25 | 92 | **2.74** | 96 |
| | WaNet | 0.41 | 36 | 0.27 | 28 | 0.57 | 28 | 0.64 | 32 | 0.54 | 34 | **0.12** | 100 |
| | IAD | 2.10 | 22 | 4.56 | 18 | 4.05 | 26 | 30.48 | 24 | 12.70 | 32 | **1.02** | 100 |
| | BppAttack | 1.76 | 18 | 1.93 | 16 | **1.06** | 24 | 8.95 | 12 | 4.75 | 18 | 1.64 | 100 |
| ImageNet | BadNets | 0.24 | 96 | **0.07** | 92 | 0.41 | 84 | 4.31 | 92 | 3.62 | 92 | 0.24 | **98** |
| | Blended | 0.34 | **100** | **0.15** | 96 | 0.28 | 96 | 1.07 | 88 | 1.26 | 96 | 0.33 | 100 |
| | WaNet | 0.29 | 20 | 0.61 | 14 | 0.45 | 34 | 0.85 | 34 | 0.57 | 28 | **0.17** | 100 |
| | IAD | 0.34 | 18 | 0.32 | 26 | 0.32 | 16 | 4.51 | 22 | **0.27** | 32 | 0.29 | 100 |
| | BppAttack | 1.91 | 4 | 2.47 | 0 | 2.85 | 36 | 4.32 | 18 | 2.17 | 42 | **1.76** | 94 |

## F.1 THE NUMBER OF BENIGN LOCAL SAMPLES

Similar to almost all existing BTI methods, our BTI-DBF also requires a few benign local samples. In our main experiments, we assume that defenders have 5% benign training samples. In this part, we evaluate the performance of our methods with different sample ratios (*i.e.*, {1%, 3%, 5%, 7%, 9%}).

As shown in Figure 6, as we expected, the performances of our methods increase with the increase of sample ratio. In particular, our methods can still reach promising performance even with only 1% benign training samples. These results verify the effectiveness of our methods.

## F.2 THE THRESHOLD OF SAMPLE DISTANCE IN THE INPUT SPACE

Recall that we need to assign a hyper-parameter $\tau$ in Eq. (2) to limit the distance between benign and poisoned samples in the input space for our BTI-DBF. We hereby discuss its effects.

As shown in Figure 7, our BTI-DBF can reach the best results for defending against all attacks when $\tau$ is set in a reasonable range (*i.e.*, $[0.2 - 0.3]$). The performance may decrease if $\tau$ is too small or too big. Defenders should assign it based on their specific needs or expertise of potential attacks.

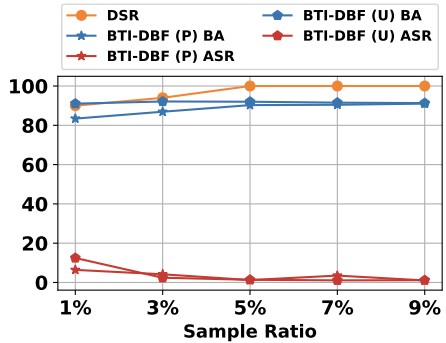
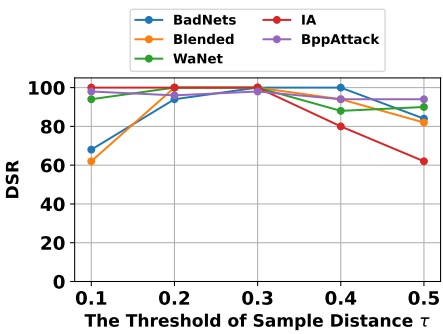

Figure 6: The effect of the number of benign local samples measured by sample ratio to the benign training samples.

Figure 7: The effect of distance threshold.

Table 11: The detection success rate (%) of our BTI-DBF with feature space defined by the output of different layers on the CIFAR-10 dataset with ResNet-18.

| Attack↓, Layer→ | 11-$th$ | 13-$th$ | 15-$th$ | 17-$th$ (Last) |
|---|---|---|---|---|
| BadNets | 94 | 100 | 100 | 100 |
| Blended | 100 | 90 | 90 | 100 |
| WaNet | 20 | 62 | 100 | 100 |
| IAD | 32 | 8 | 16 | 100 |
| BppAttack | 26 | 78 | 44 | 98 |

### F.3 THE SELECTION OF FEATURE SPACE

Recall that we define feature space as the output of the last convolution layer, following the most classical settings used in existing BTI methods. However, as demonstrated in (Jebreel et al., 2023), it may not be the best layer to differentiate between benign and poisoned samples. In this part, we explore how the selection of feature space influences the performance of our BTI-DBF.

Specifically, we define the feature space as the output of $i$-th layer and discuss its influence. As shown in Table 11, using deeper layers for trigger inversion leads to better performance in general, especially for complicated backdoor attacks (*e.g.*, WaNet and IAD). It is mostly because DNNs need more layers to 'memorize' them. Accordingly, we still suggest that users exploit the last layer as the default setting since it contains sufficient information for all attacks.

## G  ADDITIONAL RESULTS OF ABLATION STUDY

### G.1 EFFECTIVENESS OF DECOUPLING BENIGN FEATURES IN DEFENSES

In Section 4.4, we demonstrate that decoupling benign features (DBF) is critical for the performance of BTI-DBF. In this section, we verify the effectiveness of DBF in our designed backdoor defenses (*i.e.*, BTI-DBF (U) and BTI-DBF (P)).

As shown in Table 12, both BTI-DBF (U) and BTI-DBF (P) will fail in many cases without our DBF module. In particular, removing the DBF module may even decrease benign accuracy in all cases to some extent. These results verify the effectiveness of our DBF again.

### G.2 THE EFFECTIVENESS OF BENIGN LOSS FOR BTI-DBF (P)

Recall that we design a benign loss $\mathcal{L}_b$ in Eq. (6) to minimize the influence of the purification generator oo benign samples in our BTI-DBF (P). In this section, we verify its effectiveness.

Table 12: The performance (%) of designed defenses without or with decoupling benign features.

| Attack | BTI-DBF (P) | | | | BTI-DBF (U) | | | |
|---|---|---|---|---|---|---|---|---|
| | w/o DBF | | w/ DBF | | w/o DBF | | w/ DBF | |
| | BA | ASR | BA | ASR | BA | ASR | BA | ASR |
| BadNets | 88.13 | 21.66 | **90.28** | **1.23** | 91.35 | 35.08 | **92.00** | **1.36** |
| Blended | 88.90 | 6.87 | **89.13** | **1.00** | 92.51 | 18.58 | 91.60 | **7.92** |
| WaNet | 88.26 | 16.57 | **89.14** | **1.60** | 91.31 | 28.39 | 90.82 | **0.94** |
| IAD | 88.61 | 8.62 | **90.21** | **3.37** | 91.65 | 33.87 | **91.91** | **1.22** |
| LC | 90.33 | 8.75 | **90.48** | **4.15** | 91.72 | 16.50 | 90.02 | **1.11** |
| BppAttack | 89.19 | 3.71 | **89.39** | **2.52** | 92.58 | 84.44 | 90.98 | **5.02** |

Table 13: The performance (%) of our designed defenses, $i.e.$, BTI-DBF (P) and BTI-DBF (U), under all-to-all attacks on the CIFAR-10 dataset.

| | No Defense | | BTI-DBF (P) | | BTI-DBF (U) | |
|---|---|---|---|---|---|---|
| | BA | ASR | BA | ASR | BA | ASR |
| BadNets-A | 92.97 | 82.46 | 88.24 | 5.72 | 90.29 | 2.81 |
| Blended-A | 92.41 | 64.76 | 89.72 | 4.85 | 90.01 | 1.74 |
| WaNet-A | 94.76 | 86.63 | 89.94 | 4.39 | 91.17 | 1.04 |
| IAD-A | 94.12 | 92.09 | 90.03 | 0.85 | 90.83 | 1.48 |
| BppAttack-A | 85.62 | 82.04 | 81.49 | 0.21 | 83.30 | 1.57 |

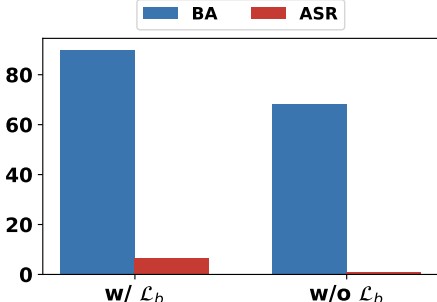

Figure 8: Effectiveness of the benign loss $\mathcal{L}_b$.

Specifically, we conduct experiments on CIFAR-10 under BadNets for discussions. As shown in Figure 8, removing our benign loss will significantly reduce benign accuracy. It is mostly because the purification generator intends to distort the whole image to a large extent without trying to learn and preserve benign features in this case. These results verify the effectiveness of our benign loss.

## H DEFENSE EVALUATION UNDER ALL-TO-ALL ATTACKS

In Section 4.3, we evaluate defenses only under the all-to-one attacks. In this section, we evaluate them under the all-to-all attacks. We conduct experiments on CIFAR-10 and extend all evaluated all-to-one attacks to the all-to-all setting with $G_Y(y) = (y+1) \mod K$. Unless otherwise specified, all settings are the same as those stated in Section 4.1.

As shown in Table 13, our defenses are still highly effective in reducing the threats of all all-to-all baseline attacks. These results confirm the effectiveness and universality of our methods.

## I MORE DETAILS ABOUT POTENTIAL ADAPTIVE ATTACKS

In this section, we provide more details of our designed potential adaptive attacks used in Section 4.5.

**Adap-Blended.** As we mentioned in Section 4.5, our methods also implicitly rely on the latent separation assumption of backdoor attacks to some extent. Accordingly, they could be bypassed by adaptive attacks that can reduce latent separation between benign and poisoned samples. Adap-Blended (Qi et al., 2023a) is currently the most advanced attack for diminishing latent separation. Specifically, it correctly label a few modified sample instead of marking them as the target label.

**Adaptive BadNets.** Recall that the first step of our BTI-DBF relies on successfully decoupling benign features. Accordingly, we can design an adaptive method based on BadNets (dubbed 'Adaptive BadNets') to directly target our BTI-DBF, where we require that using backdoor features can also correctly classify benign samples. Specifically, we pre-train a backdoored model $f_{\boldsymbol{w}}$ with BadNets at first. Then, we use BTI-DBF to decouple benign features. Finally, we fine-tune the attacked model to ensure that the remaining backdoor features can correctly classify benign samples. In conclusion, the optimization process of Adaptive BadNests can be denoted as follows:

$$\min_{\boldsymbol{w}} \sum_{(\boldsymbol{x},y)\in\mathcal{D}_l} \mathcal{L}_n + \mathcal{L}_c + \mathcal{L}_t, \tag{7}$$

where

$$\mathcal{L}_n = \mathcal{L}(f_{\boldsymbol{w}}(\boldsymbol{x}), y) + \mathcal{L}(f_{\boldsymbol{w}}(\boldsymbol{x}'), y'), \tag{8}$$

$$\mathcal{L}_c = \mathcal{L}(S_b \circ (S_a(\boldsymbol{x}) \odot \boldsymbol{m}), \, y), \tag{9}$$

$$\mathcal{L}_t = \mathcal{L}(S_b \circ (S_a(\boldsymbol{x}) \odot (1 - \boldsymbol{m})), \, y) + \mathcal{L}(S_b \circ (S_a(\boldsymbol{x}') \odot (1 - \boldsymbol{m})), \, y'). \tag{10}$$

$\mathcal{L}_n$ is used to keep $f_{\boldsymbol{w}}$ backdoored; We exploit $\mathcal{L}_c$ to ensure the functionality of benign features; In particular, $\mathcal{L}_t$ is used to make backdoor features exhibit both backdoor and benign behaviors.

However, as we can see in Table 6, our methods (*i.e.*, BTI-DBF (U) and BTI-DBF (P)) are still effective in defending against adaptive attacks. We speculate that it is because obfuscating features is a highly challenging task, while our iteration-based enhancement also plays a critical role.

## J  VISUALIZATION OF BACKDOOR TRIGGER INVERSION

In this section, we visualize the generated poisoned samples and their ground-truth version to further analyze the reliability of BTI methods. We adopt CIFAR-10 as an example for discussions.

As shown in Figure 9, only our BTI-DBF can reliably recover the shape of trigger patterns, especially those of the IAD attack. We notice that the trigger color may differ since DNNs intend to learn color parts instead of the specific color due to trigger generalizability (Cheng et al., 2021).

## K  DISCUSSIONS ABOUT EXPLOITED DATA

In this paper, all exploited samples are from open-sourced datasets (*i.e.*, CIFAR-10, GTSRB, and ImageNet). The ImageNet dataset may contain a few human objects. However, our work treats all objects the same and does not intentionally exploit or manipulate human-related content. Our modifications are also non-offensive and non-semantic. As such, our research fulfills the requirements of those datasets and should not be regarded as a violation of personal privacy.

## L  REPRODUCIBILITY STATEMENT

The appendix provides detailed information on the datasets, models, training and evaluation settings, and computational facilities. The codes and model checkpoints for reproducing our main evaluation results are also provided in the supplementary material. We will release the training codes of our methods upon the acceptance of this paper.

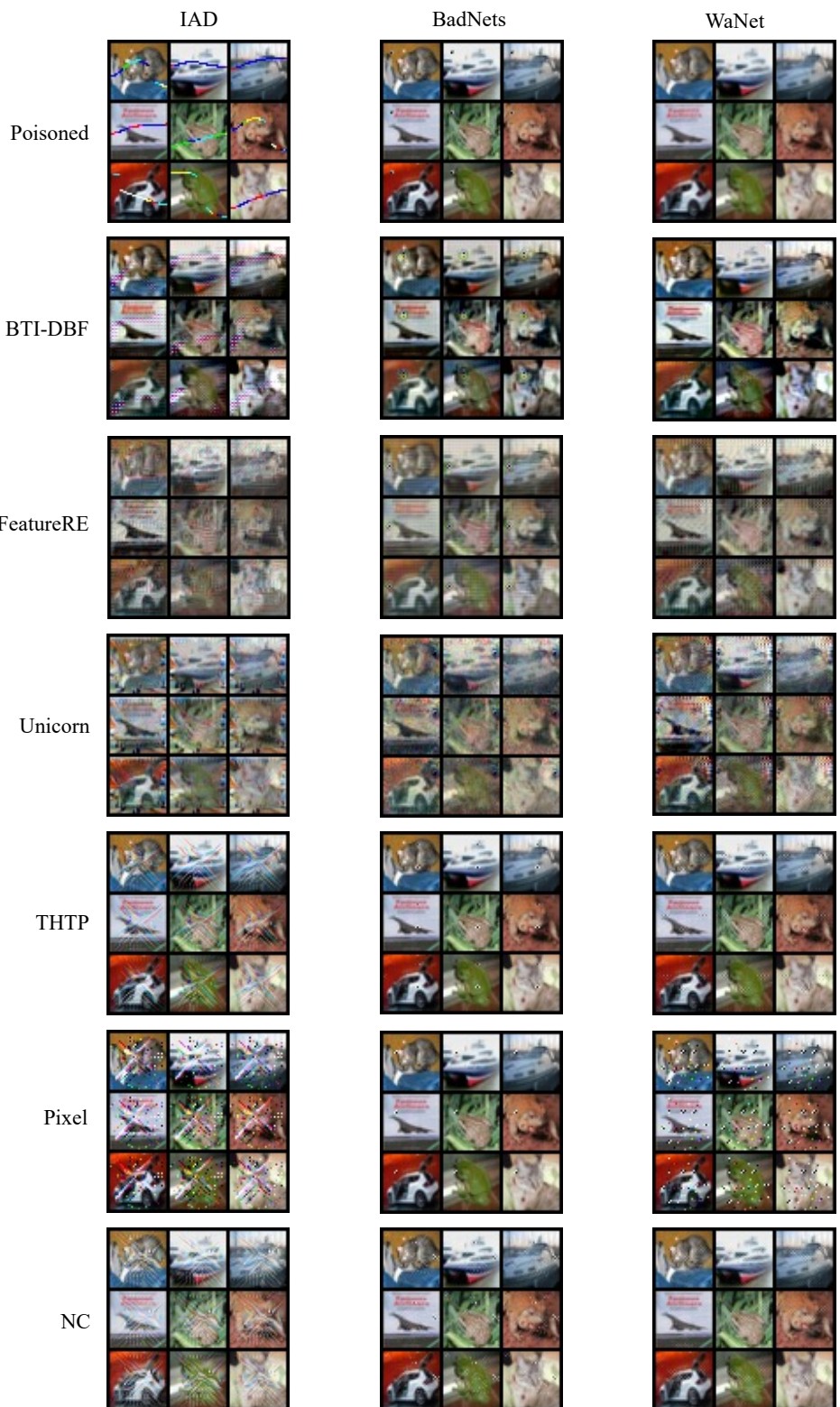

Figure 9: The example of poisoned samples and their recovered versions generated by different BTI methods on CIFAR-10. Only our BTI-DBF can reliably recover the shape of all trigger patterns.

## M    ADDITIONAL RESULTS ON THE FULL IMAGENET-1K DATASET

**Settings.** In this section, we further validate our designed defense methods on the full ImageNet-1K dataset. We set the poison rate to 5% and finetune the pre-trained ResNet-18 on ImageNet-1K to implant backdoors. We use BadNets and WaNet as attack baselines. And the triggers are the same as which we used for the subset of ImageNet.

**Results.** As shown in Table 14, both BTI-DBF (U) and BTI-DBF (P) are still effective (BA drops less than 3%, while the ASR is within 5%), even if the number of classes is huge.

Table 14: The performance (%) of our methods in defending against BadNets and WaNet on ImageNet-1K.

| Defense→ | No Defense | | BTI-DBF (P) | | BTI-DBF (U) | |
|---|---|---|---|---|---|---|
| Attack↓ , Metric→ | BA | ASR | BA | ASR | BA | ASR |
| BadNets | 59.79 | 99.63 | 57.42 | 3.57 | 57.82 | 4.16 |
| WaNet | 61.20 | 96.13 | 59.39 | 1.05 | 58.46 | 2.84 |

## N    POTENTIAL LIMITATIONS OF OUR WORK

In this section, we discuss the potential limitations of our work.

Firstly, as illustrated in Section 3.1, our defense mainly focuses on using third-party pre-trained models. In particular, similar to existing baseline methods, we assume that defenders have a few local benign samples. Accordingly, our method is not feasible without benign samples. Besides, we need to train a model for the scenarios using third-party datasets before conducting trigger inversion and follow-up defenses, which is computation- and time-consuming. We will further explore how to conduct BTI under few/zero-shot settings in our future works.

Secondly, our method needs to obtain the feature layer of the backdoored model to decouple the benign features and inverse the backdoor triggers. Besides, the optimization process in our method also relies on a white-box setting. Accordingly, it does not apply to black-box scenarios in which the defenders can only access the final output of the backdoored model. We will continue the exploration of designing black-box BTIs in our future works.

## O    DIFFERENT DISTANCE MEASUREMENT

**Settings.** In this section, we further validate our designed defense methods on the different distance measurement. We conduct experiments with $L1$ instead of $L2$ norm as our distance measurement on CIFAR-10 and ResNet-18. Other settings are the same as we used in $L2$ norm.

**Results.** As shown in table15, our methods are still highly effective under the new measurement (BA drops less than 6%, while the ASR is within 6%).

Table 15: The performance (%) of our methods in defending against six attacks when the distance measurements are L1 norm and L2 norm on CIFAR-10

| Defenses→ | No Defense | | L1 Norm | | | | L2 Norm | | | |
|---|---|---|---|---|---|---|---|---|---|---|
| | | | BTI-DBF (P) | | BTI-DBF (U) | | BTI-DBF (P) | | BTI-DBF (U) | |
| Attacks↓ , Metric→ | BA | ASR | BA | ASR | BA | ASR | BA | ASR | BA | ASR |
| BadNets | 92.82 | 99.88 | 89.80 | 1.40 | 91.09 | 0.44 | 90.28 | 1.23 | 92.00 | 1.36 |
| Blend | 93.08 | 97.31 | 88.97 | 5.14 | 88.25 | 2.48 | 89.13 | 1.00 | 91.60 | 7.92 |
| WaNet | 94.53 | 99.59 | 89.62 | 5.32 | 89.48 | 2.70 | 89.14 | 1.60 | 90.82 | 0.94 |
| IAD | 94.07 | 99.41 | 89.76 | 4.64 | 90.10 | 2.14 | 90.21 | 3.73 | 91.91 | 1.22 |
| LC | 94.65 | 88.83 | 88.95 | 4.53 | 89.26 | 2.22 | 90.02 | 1.11 | 90.48 | 4.51 |
| BppAttack | 93.88 | 99.99 | 89.57 | 1.35 | 91.76 | 4.46 | 89.39 | 2.52 | 90.98 | 5.02 |

Table 16: The performance (%) of our methods in defending against natural backdoor attack on the MeGlass dataset.

| Defenses↓ , Metric→ | BA | ASR |
|---|---|---|
| No Defense | 82.53 | 99.96 |
| BTI-DBF (P) | 75.54 | 2.70 |
| BTI-DBF (U) | 73.92 | 2.86 |

Table 17: The performance (%) of our methods in defending against the composite attack on the CIFAR-10 dataset.

| Defenses↓ , Metric→ | BA | ASR |
|---|---|---|
| No Defense | 92.79 | 97.10 |
| BTI-DBF (P) | 89.48 | 4.50 |
| BTI-DBF (U) | 90.63 | 1.88 |

Table 18: The performance (%) of our methods and NONE in defending against BadNets with different trigger size on the CIFAR-10 dataset.

| Trigger Size→ | $7 \times 7$ | | $9 \times 9$ | | $15 \times 15$ | |
|---|---|---|---|---|---|---|
| Defenses↓ , Metric→ | BA | ASR | BA | ASR | BA | ASR |
| No Defense | 92.32 | 100.00 | 92.24 | 100.00 | 92.06 | 100.00 |
| NONE | 90.28 | 1.69 | **90.54** | **1.97** | 89.78 | 23.26 |
| BTI-DBF (P) | 89.57 | 2.21 | 89.02 | 4.57 | 88.45 | 4.97 |
| BTI-DBF (U) | **90.46** | **1.07** | 90.17 | 2.36 | **90.27** | **0.86** |

# P  THE RESISTANCE TO MORE ADAPTIVE ATTACKS

## P.1  THE RESISTANCE TO NATURAL BACKDOOR ATTACK

**Settings.** In this section, we further validate our methods in defending against the natural backdoor attack (Wenger et al., 2022). We conduct experiments on MeGlass with ResNet-18. This dataset consists of 47,917 face samples of 1,710 different people. We use 38,334 (80%) samples of this dataset as our training set and others as the testing set. In this paper, we resize all images to the size of $3 \times 120 \times 120$. We use 95% of the training set to train the backdoored model and 5% for defense.

**Results.** As shown in the Table 16, our methods are still highly effective in defending against this special attack (BA drops less than 9%, while the ASR is within 3%). It is mostly because their backdoor features are still decoupled from benign features, and their triggers are still a small part of the whole image. We will further explore it in our future works.

## P.2  THE RESISTANCE TO COMPOSITE ATTACK

**Settings.** We reproduce the composite attack method with its official code under the default settings (Lin et al., 2020). We conduct experiments on CIFAR-10 with ResNet-18.

**Results.** As shown in the Table 17, both BTI-DBF (U) and BTI-DBF (P) are still effective in defending against the composite attack (BA drops less than 4%, while the ASR is within 5%). It is mostly because we use a soft mask $m \in [0, 1]$, and the benign features can still be decoupled from the backdoor features, even though they are mixed to some extent.

# Q  ADDITIONAL COMPARISONS TO NONE

**Settings.** We reproduce NONE based on its official code under the default setting (Wang et al., 2022a). We conduct experiments on the CIFAR-10 dataset with ResNet-18. Specifically, we use BadNets with $7 \times 7$, $9 \times 9$, and $15 \times 15$ three different sizes for discussions.

**Results.** As shown in the Table 18, our method is comparable with NONE under the trigger size in $7 \times 7$ and $9 \times 9$. However, we note that our defense is still effective when the trigger size is $15 \times 15$ while NONE does not work well.

# R  RESULTS UNDER ATTACKS WITH THE CHESSBOARD TRIGGER

**Settings.** We conduct experiments on the CIFAR-10 dataset with ResNet-18. We follow the default settings used in its original paper (Xiang et al., 2022) and create a global chessboard pattern, which were also used in (Wang et al., 2024b).

Table 19: The performance (%) of our methods in defending against BadNets with the 'chessboard' trigger on the CIFAR-10 dataset.

| Defenses↓, Metric→ | BA | ASR |
|---|---|---|
| No Defense | 92.56 | 100.00 |
| BTI-DBF (P) | 89.19 | 2.07 |
| BTI-DBF (U) | 90.51 | 0.00 |

**Results.** As shown in the Table 19, our defenses are still highly effective under the chessboard-type trigger (BA drops less than 4%, while the ASR is within 3%).

