# OpenReview forum: "Towards Reliable and Efficient Backdoor Trigger Inversion via Decoupling Benign Features"
_ICLR.cc/2024/Conference — ICLR 2024 spotlight_

### Official Review · Reviewer_tkgj · 2023-10-26

**Soundness:** 3 good
**Presentation:** 4 excellent
**Contribution:** 3 good
**Rating:** 8
**Confidence:** 5

**Summary:**

In this paper, a trigger inversion approach is proposed by first decoupling the benign features from the backdoor features. Then the trigger is inverted on the backdoor features. The proposed method is evaluated on several datasets compared with several baseline approaches against several popular backdoor attacks.

**Strengths:**

* The paper is generally well-organized.

**Weaknesses:**

* (Claimed contribution 3) The proposed BTI-DBF is almost the same as the backdoor mitigation approach in Neural Cleanse [1]!

I didn't flag for ethics review for this one since I tend to believe that the authors just omitted this existing approach.

[1] Wang et al,  Neural cleanse: Identifying and mitigating backdoor attacks in neural networks. In IEEE S&P, 2019.

* (Claimed contribution 2) The general idea of first decoupling backdoor features from benign ones and then performing trigger inversion on backdoor features is the same as in [2] (though the formulation of the optimization problem is different).

[2] Liu et al, ABS: Scanning Neural Networks for Back-doors by Artificial Brain Stimulation. CCS, 2019.

* (Claimed contribution 1) "Revealing the low efficiency and low similarity nature of existing backdoor trigger inversion (BTI) methods and their intrinsic reason" cannot be regarded as a contribution even though you show your method performs better. Besides, there is no adequate discussion about the "intrinsic reason" in this paper.

* The motivation of this work is weak.

What is the motivation for proposing this trigger inversion approach? If the purpose is for better backdoor detection, there is no detection performance demonstrated in the paper. If it is for better backdoor mitigation, there is no evidence that the trigger inverted by other baselines cannot mitigate the backdoor. Moreover, intuitively, inaccurately estimated triggers will introduce more robustness to backdoor unlearning. For example, if the trigger is a 3 by 3 yellow square, unlearning using yellow squares with different shapes and sizes will be more effective than unlearning the backdoor using the exact 3 by 3 square only.

* The results in Table 1 need to be double-checked.

For example, the DSR for Unicorn is much lower than the original paper [3].

[3] Wang et al, Unicorn: A unified backdoor trigger inversion framework. In ICLR, 2023.

* The intuition behind the proposed method does not always hold.

The proposed trigger inversion method can be defeated when there is no decoupling between benign and backdoor features. This happens when the model is compact and when the trigger is globally wide. For example, the "chessboard" trigger that cannot be mitigated by the method in [4] does not satisfy the decoupling assumption.

[4] Wang et al, MM-BD: Post-Training Detection of Backdoor Attacks with Arbitrary Backdoor Pattern Types Using a Maximum Margin Statistic, In IEEE S&P, 2024.

* Insufficient evaluation of the decoupling method.

If the decoupling method works for the proposed formulation for trigger inversion, it should also work for other formulations such as Unicorn. It is important to show that such decoupling is generalizable.

* No evaluation of efficiency in the main paper.

To show that the proposed method is reliable and efficient, it is necessary to include a quantitative comparison of computational overhead in the main paper.

**Questions:**

Please see the weakness part.

---

> ### Author Response · Authors · 2023-11-20
> **Author Response (Part I)**
>
> Dear Reviewer tkgj, thank you very much for your careful review of our paper and thoughtful comments. We are encouraged by your positive comments on our **good paper presentation**. We are deeply sorry that our previous submission may lead you to some misunderstandings. We hope the following responses could help clarify misunderstandings and alleviate your concerns.
>
> ---
> **Q1**: The proposed BTI-DBF is almost the same as the backdoor mitigation approach in Neural Cleanse.
>
> **R1**: Thank you for this comment! We are deeply sorry that our previous submission may lead you to some misunderstandings that we want to clarify here.
>
> - We speculate that you have this misunderstanding mainly because we all have a trigger inversion process before conducting further defenses. However, **this is the characteristic of all backdoor trigger inversion (BTI)-based methods**. We argue that it is unfair to claim our method is almost the same as neural cleanse simply based on it.
> - **Our BTI is quite different from neural cleanse in trigger inversion**. Neural cleanse used $x' = (1-m)\cdot t + m\cdot x$ to generate backdoor samples. Their $m$ and $t$ are fixed after training. However, **our method can generate different triggers via the UNet generator.** Moreover, Neural Cleanse only considers the backdoor behavior in pixel space. Our method further considers the backdoor behavior in feature space and finds the difference between the benign features and the backdoor features. Therefore, **our method can generate higher-quality triggers**. As shown in the Table in our paper, our defenses can succeed even when the triggers are dynamic, but neural cleanse can't.
> - **Our work considers more defense variants.** We apply our BTI in backdoor-removal defenses, pre-processing-based defenses, and detection-based defenses. However, **neural cleanse did not consider the pre-processing-based variant**. We have made certain breakthroughs in this regard.
>
> ---
>
> **Q2**: The general idea of first decoupling backdoor features from benign ones and then performing trigger inversion on backdoor features is the same as in [ABS: Scanning Neural Networks for Back-doors by Artificial Brain Stimulation. CCS, 2019.] (though the formulation of the optimization problem is different).
>
> **R2**: Thank you for this comment! We are deeply sorry that our previous submission may lead you to some misunderstandings that we want to clarify here.
>
> - **We never claimed that we were the first to work on backdoor trigger inversion by separating benign and backdoor features**. In our paper, we intended to emphasize that all existing methods (including ABS) first fitted backdoor features before decoupling backdoor and benign features. In contrast, our method first fits benign features, avoiding the assumption of backdoor generation.
> - Similar to neural cleanse, **ABS also assumed that backdoor features can be completely separated from benign ones at the neuron-level**. This method ignored the complex interactions between neurons. In contrast, we exploit a soft feature mask whose elements are from $[0, 1]$ instead of $\{0, 1\}$ to learn interaction effects.
>
> ---
>
> **Q3**: "Revealing the low efficiency and low similarity nature of existing backdoor trigger inversion (BTI) methods and their intrinsic reason" cannot be regarded as a contribution even though you show your method performs better. Besides, there is no adequate discussion about the "intrinsic reason" in this paper.
>
> **R3**: Thanks for your comments. We explained why exsiting BTI methods can't decouple the backdoor features in our Section 1. We are deeply sorry that we failed to describe it more clearly and lead you to some misunderstandings. We hereby provide more details about it.
> - We argue that revealing the common underlying limitations of an important defense paradigm (i.e., backdoor trigger inversion) is also an important contribution. This can alert the field against blind optimism about such methods.
> - **Intrinsic Reasons**: We reveal that both limitations are all because existing methods need to approximate and decouple backdoor features at first to separate benign and backdoor features, as required by BTI. Specifically, **these methods need to 'scan' all potential classes to speculate the target label since defenders have no prior knowledge about attacks and poisoned samples**. These processes are time-consuming since each scan requires iteratively solving a particular optimization problem.
>
> ---

---

> > ### Author Response · Authors · 2023-11-20
> > **Author Response (Part II)**
> >
> > ---
> >
> > **Q4**: The motivation of this work is weak. What is the motivation for proposing this trigger inversion approach? If the purpose is for better backdoor detection, there is no detection performance demonstrated in the paper. If it is for better backdoor mitigation, there is no evidence that the trigger inverted by other baselines cannot mitigate the backdoor. Moreover, intuitively, inaccurately estimated triggers will introduce more robustness to backdoor unlearning. For example, if the trigger is a 3 by 3 yellow square, unlearning using yellow squares with different shapes and sizes will be more effective than unlearning the backdoor using the exact 3 by 3 square only.
> >
> > **R4**: Thank you for these insightful comments! We are deeply sorry that we failed to explain why having a more accurate trigger inversion is important.
> > - **Accurate trigger inversion yields to better defenses**. As shown in Table 2 of our main paper, all baseline backdoor trigger inversion-based defenses suffer from low effectiveness in some cases, whereas our methods do not.
> > - **Accurate trigger inversion yields to better interpretability**. In addition to ultimately reducing the backdoor threat, explaining the source of the backdoor in the model is also an important requirement.
> > - **Accurate trigger inversion is the cornerstone of designing effective pre-processing-based backdoor defenses**. Otherwise, it will only have a limited effect like Februus.
> > - **Inaccurately estimated triggers would not necessarily introduce more robustness to backdoor unlearning**. As shown in Table 1 of our main paper, existing methods cannot even accurately find the target class in many cases.  It is difficult to guarantee the effectiveness of doing unlearning in this situation. Otherwise, we might as well randomize a pattern and category for unlearning.
> >
> >
> > ---
> >
> > **Q5**: The results in Table 1 need to be double-checked.
> > For example, the DSR for Unicorn is much lower than the original paper 'Unicorn: A unified backdoor trigger inversion framework'.
> >
> > **R5**: Thank you for your comment! We are deeply sorry that our metric setting may lead you to some misunderstandings. **DSR indicates whether a BTI method can find the target label correctly**. This metric is different from ASR, which is the attack success rate after defenses, and **is not used in Unicorn**.
> >
> > ---
> >
> > **Q6**: The intuition behind the proposed method does not always hold. The proposed trigger inversion method can be defeated when there is no decoupling between benign and backdoor features. This happens when the model is compact and when the trigger is globally wide. For example, the "chessboard" trigger that cannot be mitigated by the method in [4] does not satisfy the decoupling assumption.
> >
> > [4] Wang et al, MM-BD: Post-Training Detection of Backdoor Attacks with Arbitrary Backdoor Pattern Types Using a Maximum Margin Statistic, In IEEE S&P, 2024.
> >
> > **R6**: Thank you for this insightful comment! We are deeply sorry that our submission may lead you to some misunderstandings that we want to clarify here.
> >
> > - Different from existing BTI methods, **we did not assume that backdoor features can be completely separated from benign ones at the neuron-level**. This method ignored the complex interactions between neurons. In contrast, we exploit a soft feature mask whose elements are from $[0, 1]$ instead of $\{0, 1\}$ to learn interaction effects.
> > - In general, **we decouple backdoor and benign features instead of backdoor and benign pixels**. Accordingly, our method should still be effective when the trigger is globally wide in the input space.
> > - To further alleviate your concerns, following your suggestions, we conduct additional experiments with chessboard-type trigger patterns. As shown in the following table, **our defenses are still highly effective under the chessboard-type trigger**.
> >
> >
> > **Table 1. The performance (%) of our methods in defending against the "chessboard" trigger on CIFAR-10.**
> > |       | BA  | ASR  |
> > | ----  | ----  | ----  |
> > |  No Defense  | 92.56 | 100.00 |
> > | BTI-DBF (P\)  | 89.19 |  2.07 |
> > | BTI-DBF (U)  | 90.51 | 0.00  |
> >
> >
> > ---
> >
> > **Q7**: Insufficient evaluation of the decoupling method. If the decoupling method works for the proposed formulation for trigger inversion, it should also work for other formulations such as Unicorn. It is important to show that such decoupling is generalizable.
> >
> > **R7**: Thank you for this insightful comment! However, **the main difference between different backdoor trigger inversion methods lies in how to decouple benign and backdoor features**.  Therefore, we can't extend our method to other existing methods, because the extended method is formally the same as ours.
> >
> >
> > ---

---

> > > ### Author Response · Authors · 2023-11-20
> > > **Author Response (Part III)**
> > >
> > > ---
> > >
> > > **Q8**: No evaluation of efficiency in the main paper. To show that the proposed method is reliable and efficient, it is necessary to include a quantitative comparison of computational overhead in the main paper.
> > >
> > > **R8**: Thank you for this constructive suggestion! In our previous submission, we have already compared the computational overhead of different BTI methods in Figure 4 of our main paper. Specifically, **our method is significantly faster than all baseline methods**. For example, our BTI-DBF needs only 60 seconds for training, which is more than 20 times faster than Unicorn. It is still more than 3 times faster than the most efficient BTI baseline (i.e., Pixel). This efficiency advantage is even more pronounced in datasets with more classes (e.g., GTSRB and ImageNet).
> > >
> > > ---

---

> > > > ### Comment · Reviewer_tkgj · 2023-11-20
> > > > **Response to authors**
> > > >
> > > > I appreciate the author's efforts in rebuttal. All my concerns have been addressed. I have changed my rating accordingly.

---

> > > > > ### Author Response · Authors · 2023-11-21
> > > > > **Thank You for Your Positive Feedback!**
> > > > >
> > > > > Thank you so much for your positive feedback! It encourages us a lot.

---

### Official Review · Reviewer_gstB · 2023-10-31

**Soundness:** 3 good
**Presentation:** 4 excellent
**Contribution:** 3 good
**Rating:** 8
**Confidence:** 4

**Summary:**

This paper proposes a new backdoor trigger inversion method. Existing inversion
methods optimize the backdoor features, but this paper takes a different
approach that minimizes the feature differences between a benign image and its
triggered version. The method is efficient as it no longer requires scanning
of all classes of a model.

**Strengths:**

This is an interesting paper. Its main contribution is a trigger inversion
method for backdoor attacks. The main method is quite different from existing
ones, as it works as the "opposite" to existing ones by leveraging the benign
features rather than focusing on the trigger-related ones.

The method also overcomes the limitation of existing method that requires
scanning all output classes to select the most likely target label and class.

The paper has compared the proposed method with state-of-the-art baselines and achieved remarkable results.

The paper also discussed potential adaptive attacks, which is based on blending
the adverbial features into benign ones.

**Weaknesses:**

Besides the discussed adaptive attack that blend features, some attacks, e.g.,
the composite attack, "Composite Backdoor Attack for Deep Neural Network by
Mixing Existing Benign Features" from CCS 2020, also heavily mix benign and
malicious features. Similarly, the paper can benefit from evaluating on other baselines, e.g., NONE (NeurIPS'22).

**Questions:**

See detailed comments.

---

> ### Author Response · Authors · 2023-11-20
> **Author Response**
>
> Dear Reviewer gstB, thank you very much for your careful review of our paper and thoughtful comments. We are encouraged by your positive comments on our **novel approach**, **efficiency**, **remarkable performance**, **resistance to adaptive attacks**, and **good paper presentation**. We hope the following responses could alleviate your concerns.
>
> ---
> **Q1**: Besides the discussed adaptive attack that blend features, some attacks, e.g., the composite attack, "Composite Backdoor Attack for Deep Neural Network by Mixing Existing Benign Features" from CCS 2020, also heavily mix benign and malicious features. Similarly, the paper can benefit from evaluating on other baselines, e.g., NONE (NeurIPS'22).
>
>
> **R1**: Thank you for these constructive suggestions!
>
> - We argue that composite attack is well-suited as the adaptive attack against our defenses. We reproduce the composite attack method with its offical code under the default setting. As shown in the following Table 1, **both BTI-DBF (U) and BTI-DBF \(P\) are still effective in defending against the composite attack**. It is mostly because we use a soft mask $m \in [0, 1]$, and the benign features can still be decoupled from the backdoor features, even though they are mixed to some extent.
>
> **Table 1. The performance (%) of our methods in defending against the composite attack on CIFAR-10.**
>
> | Defenses $\downarrow$  Metric $\rightarrow$    | BA  | ASR  |
> | ----  | ----  | ----  |
> | No Defense  | 92.79 | 97.10 |
> | BTI-DBF (P\)  | 89.48 | 4.50 |
> | BTI-DBF (U)  | 90.63 | 1.88  |
>
> - Due to the time constraint, we use BadNets with different trigger sizes for discussions. We reproduce NONE based on its official code under the default setting. As shown in Table 2, our method is comparable with NONE under the trigger size in $7\times7$ and $9\times9$. However, we note that our defense is still effective when the trigger size is $15\times15$ while NONE does not work well.
>
>
>
>
> **Table 2. The performance (%) of our methods and NONE in defending against BadNets with different trigger size on CIFAR-10.**
> |  Trigger Size $\rightarrow$   | $7\times7$ | $9\times9$ | $15\times15$ |
> | ----  | ----  | ----  | ----  |
> |  Defenses $\downarrow$  Metric $\rightarrow$   | BA / ASR | BA / ASR | BA / ASR |
> |  No Defense  | 92.32 / 100.00 | 92.24 / 100.00 | 92.06 / 100.00 |
> |  NONE  | 90.28 / 1.69  | 90.54 / 1.97 | 89.78 / 23.26|
> |  BTI-DBF (P\)  |89.57 / 2.21 | 89.02 / 4.57 | 88.45 / 4.97|
> |  BTI-DBF (U)  |90.46 / 1.07 |90.17 / 2.36 |90.27 / 0.86 |
>
>
>
> Please refer to Appendix P.2 of our revision for more details.
>
>
> ---

---

> ### Author Response · Authors · 2023-11-21
> **Thanks to Reviewer gstB**
>
> Please allow us to thank you again for reviewing our paper and the valuable feedback, and in particular for recognizing the strengths of our paper in terms of *novelty*, *efficiency and effectiveness*, *resistance to adaptive attacks*, and *good writing*.
>
> Kindly let us know if our response and the new experiments have properly addressed your concerns. We are more than happy to answer any additional questions during the post-rebuttal period. Your feedback will be greatly appreciated.

---

> ### Author Response · Authors · 2023-11-22
> **A Gentle Reminder of the Final Feedback**
>
> We would like to thank the reviewer for the helpful discussion during the first round of the review. We hope our response has adequately addressed your comments related to the resistance of our methods to the composite attack and the comparison to NONE baseline. We take this as a great opportunity to improve our work and shall be grateful for any additional feedback you could give to us.

---

### Official Review · Reviewer_SUKT · 2023-10-31

**Soundness:** 3 good
**Presentation:** 3 good
**Contribution:** 3 good
**Rating:** 6
**Confidence:** 4

**Summary:**

Existing trigger inversion techniques optimizes the trigger to find malicious
features. This paper goes the other way and tries to optimize the image so that
the benign features to be close. This is a new angle of optimizing the trigger.
The evaluation is comprehensive including a lot of datasets, models, and
baseline methods. Results are promising.

**Strengths:**

The inversion technique is novel and different from existing works.

The proposed method can work as different variants on different phases of the
defense.

The evaluation is comprehensive, using different datasets and baselines, etc.

**Weaknesses:**

An intuition of existing backdoor trigger inversion method is that backdoor
feature pattern is relatively fixed and in small size, e.g., a patch or a filter
or a generative function. However, the feature space of benign samples can be
huge, for example, for the class horse, there could be so many types of benign
feature clusters. We are not sure if there is only one cluster in the feature
space or there are actually many of them. Thus, the optimization directions can
be relatively random. Have you tried different versions of benign features
(e.g., different distance measurement)?

The adaptive settings consider blending the benign and poisoned samples in the
feature space. Have you considered triggers that naturally appear in the
training dataset, i.e., natural triggers?

**Questions:**

How is the performance on natural triggers?

---

> ### Author Response · Authors · 2023-11-20
> **Author Response**
>
> Dear Reviewer SUKT, thank you very much for your careful review of our paper and thoughtful comments. We are encouraged by your positive comments on our **novel and different approach**, **different defense variants**, **comprehensive evaluation**, and **good paper presentation**. We hope the following responses could help clarify potential misunderstandings and alleviate your concerns.
>
> ---
> **Q1**: An intuition of existing backdoor trigger inversion method is that backdoor feature pattern is relatively fixed and in small size, e.g., a patch or a filter or a generative function. However, the feature space of benign samples can be huge, for example, for the class horse, there could be so many types of benign feature clusters. We are not sure if there is only one cluster in the feature space or there are actually many of them. Thus, the optimization directions can be relatively random. Have you tried different versions of benign features (e.g., different distance measurement)?
>
> **R1**: Thanks for this insightful question! We are deeply sorry that our submission may lead you to some misunderstandings that we want to clarify.
>
> - **We did not assume that backdoor features are relatively fixed and in small size to a large extent**. One of the difference between our method and existing methods is that we did not assume that poisoned samples are with the formulation $x' = (1-m) \cdot x + m\cdot t$ (with the regularization term $||m||$).
> - In our method, for the backdoor features, **we only assume that we can decouple them with benign features**. In particular, we assume that all elements in our feature mask are from $[0,1]$ instead of $\{0, 1\}$. Accordingly, our approach can truly decouple benign and backdoor features at the feature level rather than at the neuron level. It alleviate the assumption of previous methods that backdoor features were only a very small group.
> - In our trigger inversion step, we did assume that the distance between each poisoned sample and their benign version is small. However, **it is even practical even for attacks with visible triggers** (e.g., BadNets) since they only change a few pixels.
> - However, we fully understand your concerns. To alleviate them, we conduct additional experiments with $L_1$ instead of $L_2$ norm as our distance measurement. As shown in the following table, **our methods are still highly effective under the new measurement**.
>
>
>
>
>
> **Table 1. The performance (%) of our methods in defending against six attacks when the distance measurements are L1 norm and L2 norm on CIFAR-10**
>
> |  Defense $\rightarrow$   | No Defense |BTI-DBF (P\) |BTI-DBF (U) |BTI-DBF (P\) |  BTI-DBF (U) |
> | ----  | ----  | ----  |----  |  ----  |----  |
> | Attack $\downarrow$  Metric $\rightarrow$    | BA / ASR | BA / ASR | BA / ASR | BA / ASR |BA / ASR |
> | Distance Measurement $\rightarrow$  | - | L1 Norm | L1 Norm  |L2 Norm | L2 Norm  |
> | BadNets   | 92.82/99.88 | 89.80/1.40 |91.09/0.44  |90.28/1.23| 92.00/1.36
> | Blend  | 93.08/97.31 | 88.97/5.14  |88.25/2.48  |89.13/1.00|  91.60/7.92
> | WaNet  | 94.53/99.59 | 89.62/5.32 |89.48/2.70  |89.14/1.60|  90.82/0.94
> | IAD  | 94.07/99.41 | 89.76/4.64 |90.10/2.14  |90.21/3.73|  91.91/1.22
> | LC  | 94.65/88.83 | 88.95/4.53 |89.26/2.22   |90.02/1.11|  90.48/4.51
> | BppAttack | 93.88/99.99 | 89.57/1.35 |91.76/4.46  |89.39/2.52| 90.98/5.02
>
>
> We have provided more details in Appendix O of our revision.
>
> ---
>
>
>
> **Q2**: The adaptive settings consider blending the benign and poisoned samples in the feature space. Have you considered triggers that naturally appear in the training dataset, i.e., natural triggers?
>
> **R2**: Thank you for this insightful question! We have to admit that we did not think about it. Following similar settings provided in the open-sourced codes of [1], we conduct additional experiments on the MeGlass dataset. As shown in the following table, **our methods are still highly effective in defending against this special attack**. It is mostly because their backdoor features are still decoupled from benign features, and their triggers are still a small part of the whole image (please find more details in our R1). We will further explore it in our future works. Please refer to Appendix P.1 of our revision for more details.
>
> **Reference**
> 1. Emily Wenger, et al. "Finding Naturally Occurring Physical Backdoors in Image Datasets." NeurIPS, 2022.
>
> **Table 2. The performance (%) of our methods in defending against natural backdoor attack on the MeGlass dataset.**
>
> | Defenses $\downarrow$  Metric $\rightarrow$    | BA  | ASR  |
> | ----  | ----  | ----  |
> | No Defense  | 82.53 | 99.96 |
> | BTI-DBF (P\)  | 75.54 | 2.70 |
> | BTI-DBF (U)  | 73.92 | 2.86  |
>
>
>
> ---

---

> > ### Public Comment · ~Zhixiao_Wu1 · 2024-11-26
> > **Different views on the priors of poisoned features.**
> >
> > Hello, I believe the reviewer's concern mainly stems from Figure 2, which the authors use to illustrate the limitations of existing methods. In the figure, there are multiple clusters of normal data, and the distances to the poisoned data vary. The authors did not provide a similar visualization, which naturally leads us to consider the risk that the benign features selected by the authors' method might be far from the poisoned data. I understand that the authors' loss function is likely to identify appropriate benign features, but the logic itself is not entirely self-contained. The authors' priors about poisoned data are, I believe, incomplete. Currently, many advanced attack features are concealed（see paper "Backdoor Attack with Sparse and Invisible Trigger"）, such as embedding a poisoned feature in the shape of an airplane's wing within an image, which can easily be confused with the image's inherent noise, making detection difficult. What truly separates poisoned images from normal ones should be the mapping relationship between inputs and outputs, rather than the features themselves.

---

> ### Author Response · Authors · 2023-11-21
> **Thanks to Reviewer SUKT**
>
> Please allow us to thank you again for reviewing our paper and the valuable feedback, and in particular for recognizing the strengths of our paper in terms of *novelty*, *comprehensive experments*, and *good writing*.
>
> Kindly let us know if our response and the new experiments have properly addressed your concerns. We are more than happy to answer any additional questions during the post-rebuttal period. Your feedback will be greatly appreciated.

---

> ### Author Response · Authors · 2023-11-22
> **A Gentle Reminder of the Final Feedback**
>
> We would like to thank the reviewer for the helpful discussion during the first round of the review. We hope our response has adequately addressed your comments related to our designs for decoupling benign features and the resistance of our methods to the natural backdoor attack. We take this as a great opportunity to improve our work and shall be grateful for any additional feedback you could give to us.

---

> ### Author Response · Authors · 2023-11-23
> **A Second Reminder of the Post-rebuttal Feedback**
>
> Dear Reviewer SUKT,
>
> We greatly appreciate your initial comments. We totally understand that you may be extremely busy at this time. But we still hope that you could have a quick look at our responses to your concerns. We appreciate any feedback you could give to us. We also hope that you could kindly update the rating if your questions have been addressed. We are also happy to answer any additional questions before the rebuttal ends.
>
> Best Regards,
>
> Paper2275 Authors

---

### Official Review · Reviewer_pvdL · 2023-11-01

**Soundness:** 3 good
**Presentation:** 3 good
**Contribution:** 3 good
**Rating:** 8
**Confidence:** 4

**Summary:**

In the context of backdoor, this paper delves into the challenges in Backdoor Trigger Inversion (BTI), a critical method in defending against these threats.

Traditional BTI methods have been hindered by their reliance on extracting backdoor features without prior knowledge about the adversaries' trigger patterns or target labels, leading to suboptimal performance.

The authors propose a novel approach that inverts this paradigm by focusing on the decoupling of benign features (rather than backdoored features), followed by a refined trigger inversion process. This two-step method not only enhances the efficiency by obviating the need to scan all classes for potential target labels but also improves detection accuracy.

The paper's methodology encompasses minimizing the disparities between benign samples and their generated poisoned counterparts in the benign feature space, while maximizing differences in the backdoor features.

This approach also lays the groundwork for more effective backdoor-removal and pre-processing-based defenses.
The effectiveness of this new method is demonstrated through extensive experiments on benchmark datasets, where it achieves state-of-the-art performance in mitigating backdoor threats, showcasing a significant advancement.

**Strengths:**

- The paper proposes a novel approach to conduct trigger inversion, which is insightful. The approach is intuitive and appears effective.
- The paper provides a comprehensive evaluation to show the effectiveness and efficiency.

**Weaknesses:**

- No discussion on the limitations.

**Questions:**

1. Table 8 shows that the evaluation only picks 100 classes from ImageNet. This is wired. Has the method been tested on 1000 classes? What is the scalability of the proposed method? How does the method perform compared to other methods when the number of classes increases?

2. Section 2.2 misses some latest work on feature level BTI:
- SSL-Cleanse: Trojan Detection and Mitigation in Self-Supervised Learning, M. Zheng et al., 2023
- Detecting Backdoors in Pre-trained Encoders, S. Feng et al., CVPR'2023

Although these 2 works focus on self-supervised learning, they are highly related to the feature level BTI. It would be better to discuss them in Section 2.2.

---

> ### Author Response · Authors · 2023-11-20
> **Author Response**
>
> Dear Reviewer pvdL, thank you very much for your careful review of our paper and thoughtful comments. We are encouraged by your positive comments on our **insightful and novel method**, **effectiveness**, **efficiency**, **comprehensive evaluation**, and **good paper presentation**. We hope the following responses could help clarify potential misunderstandings and alleviate your concerns.
>
> ---
> **Q1**: Table 8 shows that the evaluation only picks 100 classes from ImageNet. This is wired. Has the method been tested on 1000 classes? What is the scalability of the proposed method? How does the method perform compared to other methods when the number of classes increases?
>
> **R1**: Thank you for the insightful question! We are deeply sorry that our dataset selection may lead you to concerns about our effectiveness.
>
> - **Using ImageNet subset is a common setting in existing backdoor-related works** (e.g., [1, 2]). For example, Fu et al. [1] picked 20 classes and Huang et al. [2] picked 30 classes from ImageNet. In our work, we deliberately chose a relatively large number of categories (i.e., 100) to evaluate the scalability of all methods.
> - However, we do understand your concerns. To further alleviate them, we conduct additional experiments on the ImageNet-1K dataset. Due to the time constraint, we use BadNets and WaNet as attack baselines. Specifically, we set the poison rate to 5\% and finetune the pre-trained ResNet-18 on ImageNet-1k to implant backdoors. As shown in Table 1, **both BTI-DBF (U) and BTI-DBF \(P\) are still effective** (BA drops less than 3%, while the ASR is within 5%), even if the number of classes is huge.
>
>
> We have provided more details in Appendix M of our revision.
>
>
> **Reference**
> 1. Kunzhe Huang, et al. "Backdoor Defense via Decoupling the Training Process." ICLR 2022.
> 2. Chong Fu, et al. "FreeEagle: Detecting Complex Neural Trojans in Data-Free Cases." USENIX Security 2023.
>
>
>
> **Table 1. The performance (%) of our methods in defending against BadNets and WaNet on ImageNet-1K.**
> |  Defense $\rightarrow$   |No Defense  | BTI-DBF (P\)| BTI-DBF (U)|
> | ----  | ----  | ----  | ----  |
> | Attack $\downarrow$  Metric $\rightarrow$  |BA / ASR  | BA / ASR| BA / ASR|
> | BadNets  | 59.79 / 99.63 |57.82 / 4.16|57.42 / 3.57|
> | WaNet  | 61.20 / 96.13 |58.46 / 2.84|59.39 / 1.05|
>
> ---
>
> **Q2**: Section 2.2 misses some latest work on feature level BTI：**(1)** SSL-Cleanse: Trojan Detection and Mitigation in Self-Supervised Learning, 2023. **(2)** Detecting Backdoors in Pre-trained Encoders, 2023. Although these 2 works focus on self-supervised learning, they are highly related to the feature level BTI. It would be better to discuss them in Section 2.2.
>
>
> **R2**: Thank you for this constructive suggestion! After carefully reading their papers, we have added some discussions in Section 2.2, as follows: **Besides, two recent research discussed how to conduct BTI in the feature space under self-supervised learning, inspired by neural cleanse**. Besides, we summarize the differences between their works and our method as follows.
>
>
> - **Motivation:** These two works focused on the BTI in self-supervised Learning in which input labels are not available. However, our work primarily address the limitations in low efficiency and effectiveness of existing BTI method under supervised learning.
> - **Method**. These two works adopted the same 'blended strategy' as neural cleanse in approximating poisoned samples, although the approximation is in the feature space instead of the input space. In contrast, our method only relies on benign samples without needing to assign a particular poisoning form for approximation.
>
>
> ---
>
> **Q3**: No discussion on the limitations.
>
> **R3**: Thank you for this insightful question! We are deeply sorry that we missed the discussion about potential limitations of our method. We hereby provide it, as follows.
>
> - As illustrated in Section 3.1, our defense mainly focuses on using third-party pre-trained models. In particular, similar to existing baseline methods, we assume that defenders have a few local benign samples. Accordingly, our method is not feasible without benign samples. Besides, we need to train a model for the scenarios using third-party datasets before conducting trigger inversion and follow-up defenses, which is computation- and time-consuming. We will further explore how to conduct BTI under few/zero-shot settings in our future works.
> - Our method needs to obtain the feature layer of the backdoored model to decouple the benign features and inverse the backdoor triggers. Besides, the optimization process in our method also relies on a white-box setting. Accordingly, it does not apply to black-box scenarios in which the defenders can only access the final output of the backdoored model. We will continue the exploration of the black-box BTI in our future works.
>
> We have added the disscussion in Appendix N of our revision.
>
>
> ---

---

> > ### Comment · Reviewer_pvdL · 2023-11-20
> >
> > I appreciate authors' efforts in rebuttal. Since all my concerns are well addressed, I raise the score accordingly.

---

> > > ### Author Response · Authors · 2023-11-21
> > > **Thank You for Your Positive Feedback!**
> > >
> > > Thank you so much for your positive feedback! It encourages us a lot.

---

### Meta-Review · Area_Chair_LwKE · 2023-12-12

**Metareview:**

This paper introduces an innovative trigger inversion method for detecting backdoored models, offering a fresh perspective by focusing on decoupling benign features instead of backdoor features. This approach not only enhances accuracy but also boosts efficiency by eliminating the need to examine all classes for potential target labels. The unanimous consensus among reviewers is that the method is both technically sound and relevant to the community. Key strengths highlighted include its novel approach, effectiveness, and thorough evaluation. The authors have successfully addressed all questions and concerns during the rebuttal phase, further strengthening the paper. In light of these factors, I strongly recommend acceptance of this paper.

**Justification For Why Not Higher Score:**

This paper addresses an interesting and important problem in the AI safety community. While the idea of backdoor trigger inversion is not new, this paper introduces a fresh perspective on the problem. A detailed experimental study demonstrates the effectiveness of the proposed method. The contributions presented in this paper deserve a spotlight presentation. Due to the lack of theoretical insights I do not recommend an oral presentation.

**Justification For Why Not Lower Score:**

The very positive feedback from the reviewers supports the decision of recommending a spotlight presentation. A lower score could be justified by the lack of theoretical insights, and the limited number of considered datasets.

---

### Decision · Program_Chairs · 2024-01-16

Accept (spotlight)